# Evolution and expression patterns of the neo-sex chromosomes of the crested ibis

Lulu Xu [1], Yandong Ren[1], Jiahong Wu [2], Tingting Cui[1], Rong Dong[3], Chen Huang[1], Zhe Feng[1], Tianmin Zhang[1], Peng Yang[1], Jiaqing Yuan [1], Xiao Xu[1], Jiao Liu[2], Jinhong Wang[1], Wu Chen [4], Da Mi[5], David M. Irwin [6], Yaping Yan[1], Luohao Xu [2] ✉, Xiaoping Yu [1] ✉ & Gang Li [1,4] ✉

Bird sex chromosomes play a unique role in sex-determination, and affect the sexual morphology and behavior of bird species. Core waterbirds, a major clade of birds, share the common characteristics of being sexually mono-morphic and having lower levels of inter-sexual conflict, yet their sex chromosome evolution remains poorly understood. Here, by we analyse of a chromosome-level assembly of a female crested ibis (Nipponia nippon), a typical core waterbird. We identify neo-sex chromosomes resulting from fusion of microchromosomes with ancient sex chromosomes. These fusion events likely occurred following the divergence of Threskiornithidae and Ardeidae. The neo-W chromosome of the crested ibis exhibits the character-istics of slow degradation, which is reflected in its retention of abundant gametologous genes. Neo-W chromosome genes display an apparent ovary-biased gene expression, which is largely driven by genes that are retained on the crested ibis W chromosome but lost in other bird species. These results provide new insights into the evolutionary history and expression patterns for the sex chromosomes of bird species.

Sex chromosomes have long been a research focus due to their unique transmission patterns and atypical structure[1,2]. In most birds and mammals, sex is determined by a pair of heteromorphic sex chromosomes (Z–W in birds and X–Y in mammals) that differ in morphology, size, and gene content[3,4]. In most birds and mammals, the Z and X chromosomes, respectively, are evolutionarily conserved, with relatively stable structures and gene contents[5,6], while the sex-limited W and Y chromosomes exhibit a high degree of hetero-chromatinization and typically contain only a few genes due to the lack of recombination[2,7–9].

In contrast to the commonly observed large-scale degradation of mammalian Y chromosomes, degradation of bird W chromosomes varies from the primitive homomorphic sex chromosomes found in ratites[10] to the fully differentiated dimorphic sex chromosomes seen in

chickens. In addition, unlike the male-biased function of the mam-malian Y chromosomes[11], previous studies with models of core land birds (Telluraves)[12], such as the collared flycatcher (*Ficedula albicollis*)[13] and the birds of paradise (Paradisaeidae)[14], suggest that no female-biased gene expression of W-linked genes is seen in birds. Genes that have remained on the W chromosome instead may have been selected to maintain gene dosage and preserve the ancestral expression levels of essential genes, rather than being the targets for female-specific functions[13].

The evolution of sex-limited chromosomes (W or Y) demonstrates an enormous amount of species-specificity and a fast rate of change[15,16]. This process is influenced by a variety of factors, including life history, mating system, and the direction of sexual selection[17,18]. Currently, our knowledge on the evolution of bird W chromosomes is

[1]College of Life Sciences, Shaanxi Normal University, Xi'an, China. [2]MOE Key Laboratory of Freshwater Fish Reproduction and Development, School of Life Sciences, Southwest University, Chongqing, China. [3]Research Center for Qinling Giant Panda, Shaanxi Academy of Forestry, Xi'an, China. [4]Guangzhou Wildlife Research Center, Guangzhou Zoo, Guangzhou, China. [5]Xi'an Haorui Genomics Technology Co., LTD, Xi'an, China. [6]Department of Laboratory Medicine and Pathobiology, University of Toronto, Toronto, ON M5S 1A8, Canada. ✉e-mail: luohaox@gmail.com; yuxp64@163.com; gli@snnu.edu.cn

mainly derived from studies in a limited number of model species, such as ostrich[19,20], emu[21,22], chicken[23], mallard[24], and some songbirds[14,25]. Our understanding of the evolution of W chromosomes in another major bird lineage, the core waterbirds (Aequornithia), which includes pelicans, herons, ibises, cormorants, fulmars, penguins, loons and stocks[12,26–31], is poor. Core waterbirds have unique characteristics, including relatively long generation times[32–34], low evolutionary rates[35,36], and sexual monomorphic traits[37,38]. Unlike bird species with sexual dimorphisms, which are crucial for reproductive behaviors and mate choice, birds with sexually monomorphic traits experience lower levels of sexual selection[39,40] and moderate levels of inter-sexual conflict[41] or antagonism[42]. Levels of inter-sexual conflict or antagonism have been proposed to drive the evolution of sex chromosomes[43], including suppressing recombination between sex chromosomes and allowing the degradation of W chromosomes[14], although this hypothesis has been challenged[44].

To gain a better understanding of sex chromosome evolution in birds, we present analyses based on the genome of the crested ibis (*Nipponia nippon*), a typical waterbird belonging to the Threskiornithidae family of the Pelecaniformes order. This species has gained much attention due to successful conservation efforts, which allowed it to escape from the brink of extinction[45,46]. The crested ibis is a large-bodied waterbird with sex-homomorphic traits, making it difficult to distinguish between sexes based on appearance. Additionally, the crested ibis is monogamous and does not exhibit any apparent courtship behavior[47]. Male and female parents share equal investment in the parental care of offspring, including nesting, hatching, and brooding[48]. In other birds, sex differences in parental investment are believed to play a major role in determining the degree of sexual selection[49]. Characterized by the low pressure of sexual selection and negligible sexual antagonism between the two sexes, the crested ibis provides an ideal model for studying the evolution of sex chromosomes in core waterbirds.

In this work, we de novo assembled a high-quality chromosomal-level reference genome of a female crested ibis and conducted comparative genomic analyses, as well as testing the global levels of gene expression of the sex chromosomes in somatic and gonadal tissues with RNA-seq data. Our comparative genomics approach, based on the first chromosome-structured W chromosome from a representative core waterbird species, allowed us to elucidate the structure, gene composition, evolutionary history, and expression patterns of the sex chromosomes in core waterbirds. These findings provide insights into sex chromosome evolution across all birds.

## Results

### A new chromosome-level genome assembly for a female crested ibis

By leveraging our PacBio HiFi reads (~37× whole-genome coverage), short reads (~60×), and chromatin conformation capture (Hi-C) reads (~184×, Supplementary Table 1), we de novo assembled a high-quality genome for a female crested ibis, representing the first published chromosome-level genome for a core waterbird. The contig N50 reached 16.4 Mb, 630-fold larger than a previous draft genome[45]. Our assembly anchored more than 95.8% of the contigs onto 29 chromosome models (Fig. 1a), including nine macrochromosomes (chr1–9), 18 microchromosomes (chr10–27), and a pair of ZW sex chromosomes (Supplementary Fig. 1). According to the known karyotype of the crested ibis (2$n$ = 68), six chromosome models are unfortunately missing from our assembly. The missing chromosomes are probably dot-like microchromosomes that can be better resolved by ONT ultra-long reads[50]. The assembled genome has a length of 1.31 Gb, a bit larger than most short-read-based bird genomes (mean size 1.1 Gb[36]), with high continuity and completeness (BUSCO: 97.7%, Supplementary Tables 2 and 3), which is similar to other published bird genomes with top assembly qualities (Supplementary Table 2).

We identified the Z and W chromosomes by comparing the sequencing coverage between males and females. In females, both the Z and W chromosomes exhibit half the sequencing coverage of autosomes, while in males, the W chromosome displays sparse coverage (Fig. 1b). The assembled Z chromosome is the sixth largest chromosome, also consistent with a cytogenetic study[51], contains 1112 genes (Supplementary Table 3). Chromosome Z uniformly exhibits a 2-fold elevation of sequencing coverage in males relative to females (Fig. 1b), in agreement with the expectation for a differentiated Z chromosome.

The sex differentiation region (SDR) of the W chromosome is 37.7 Mb in size, being one of the largest assembled bird W chromosomes (Supplementary Data 1). Such a large size for the assembled W chromosome is consistent with the cytogenetically estimated size[51] that falls between the sizes of chromosomes 9 and 10. We annotated 414 protein-coding genes on the sexually differentiated region (SDR) of chromosome W, making it the most gene-rich SDR of bird W chromosomes. After removing genes that replicated independently on the W chromosome, 83.4% of the original gene content was lost from the W chromosome, in contrast to the greater than 90% found in most land birds[52]. The Z chromosome encompasses a moderate proportion of transposable elements (TEs) (13.8%), similar to that of autosomes, and much lower than the W chromosome (39.5%, Supplementary Fig. 2a). Overall, microchromosomes have a significantly higher gene density and GC content, but fewer TEs than macro- and sex- chromosomes (two-sided $t$-test and One-way Welch's ANOVA test, Supplementary Fig. 2b–d), consistent with studies in other birds[24,36].

### Frequent inter-chromosomal rearrangements are likely associated with TE activity

Bird genomes usually possess a conserved karyotype with typically ~10 pairs of macrochromosomes and ~30 pairs of microchromosomes (2$n$ = 80) in the course of over 100 million years of evolution[36,53]. Core waterbirds, however, frequently have a smaller number of chromosomes, with many species having a diploid chromosome number of 68, including crested ibis[51]. To investigate the karyotypical evolution of the crested ibis, we conducted a multi-way comparison of genome synteny using the chromosome-level genomes of the crested ibis (NNI) and several representative bird species, including the zebra finch (*Taeniopygia guttata*, TGU, Neoaves), California condor (*Gymnogyps californianus*, GCA, Neoaves), great bustard (*Otis tarda*, OTA, Neoaves), mallard (*Aanas platyrhynchos*, APL, Galloanserae), chicken (*Gallus gallus*, GGA, Galloanserae), and emu (*Dromaius novaehollandiae*, DNO, Palaeognathae), which cover all three major bird clades. Our analysis revealed that crested ibis experienced zero fissions but at least six fusion events (Fig. 1c), a pattern previously seen in other birds with excessive numbers of chromosomal changes[53]. Among the fusions, four were between macro- and microchromosomes, which is considered the most common form of fusion in birds[21]. Fusions between macrochromosomes are surprising as they are rare in birds, but two were observed in the crested ibis (between chromosomes 1a and 6 (chr1a+chr6) and between chromosomes 7 and 8 (chr7+chr8)). Curiously, the chromosomes of moderate chromosome size that experienced fusions in the crested ibis are frequently involved in fusions in independent parrot lineages (e.g., chromosomes 6–12 and 14 (chr6–12, chr14)), suggesting that some chromosomes are more prone to chromosome fusion events.

Next, we asked whether certain TE families are associated with chromosomal changes in the crested ibis as suggested by many other authors[53,54]. We identified a species-specific outbreak of the DNA TE family (DNA-hAT-charlie) in the crested ibis genome that is significantly enriched on chromosomes with inter-chromosomal rearrangements (Fig. 1d, e, two-sided $t$-test, $P$ = 2e-02) and are concentrated to the termini of chromosomes (Supplementary Fig. 3, Chi-Squared Test). In contrast, other TE insertions do not show a significant bias towards any specific chromosome (Fig. 1f, g, Supplementary Fig. 4, two-sided $t$-test). We estimated that the DNA-hAT-

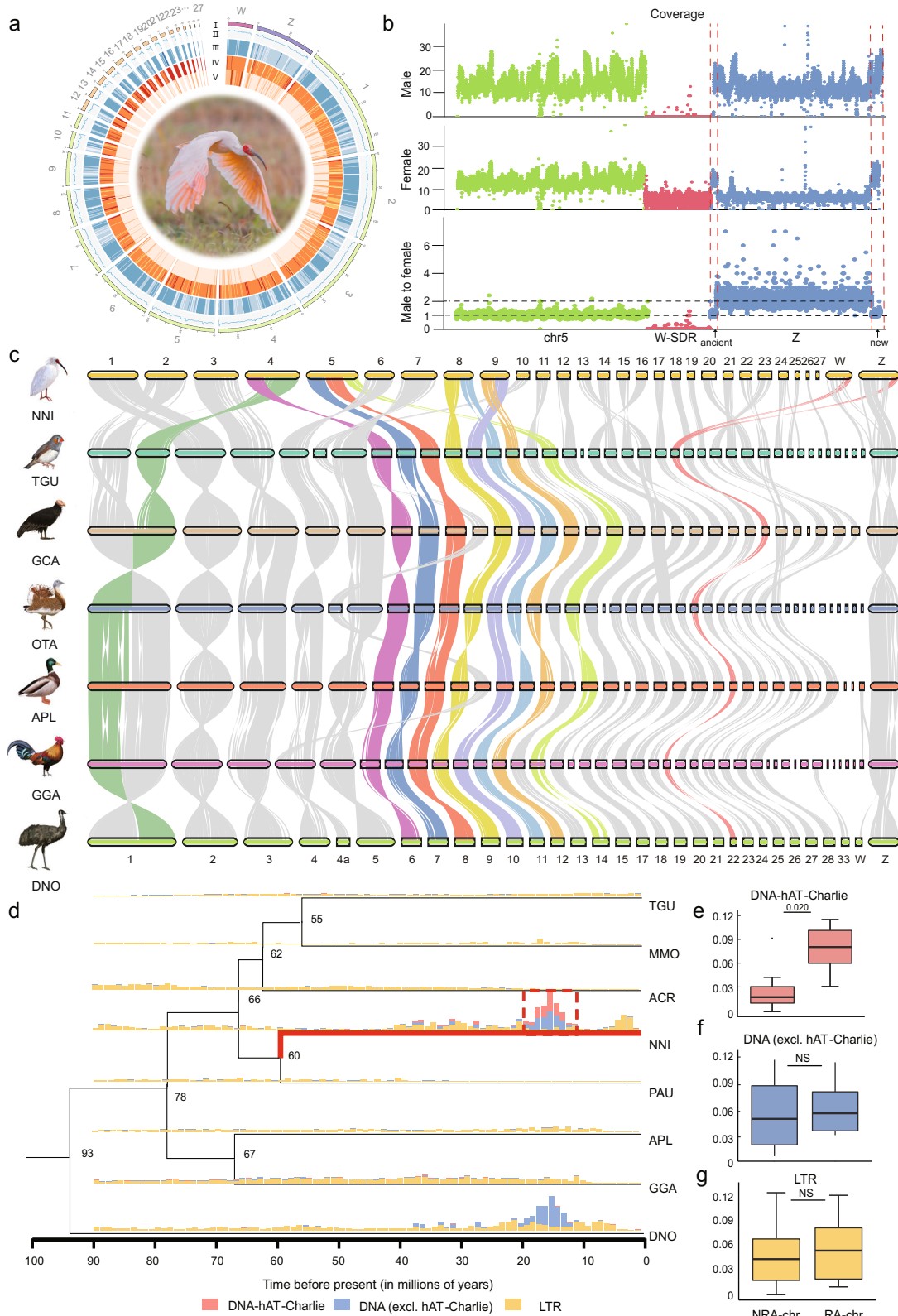

charlie elements started propagating in the crested ibis' genome approximately 15 million years ago (Fig.1d).

## Novel formation of a neo-sex chromosome pair in the crested ibis

Among the six chromosomal fusions, one appears to be an addition of chromosome 22 (chr22) to both the ancient-Z and the ancient-W (Fig. 1c). Chromosome 22 is a microchromosome and had its entire length added to the ancient-sex chromosome. This resembles the formation of the neo-sex chromosomes in parrots where an entire microchromosome (chromosome 11) joined the sex chromosome and unlike the scenario in songbirds only a part of a macrochromosome translocated to the sex chromosome[55–58]. To our knowledge, these are the first neo-sex chromosomes discovered in core waterbirds.

**Fig. 1 | Genomic comparisons between the crested ibis and other avian species.**
**a** Circos plot of the crested ibis genome. This plot displays I: karyotype; II: GC percent; III: SNP density; IV: gene density; V: repeat sequence density. **b** Illumina sequencing coverage from a male and a female over the Z and W (using chromosome 5 as a reference). Sequencing coverage was calculated per 10-kb window (dot). The arrows point to the two PARs (ancient and new-PAR). **c** Pairwise whole-genome alignments across 7 bird genomes. Chromosome IDs of crested ibis (NNI) and emu (DNO) are labeled at the top and bottom of each bar, respectively. Rearranged chromosomes are highlighted in different colors. **d** Comparison of the insertion history of TEs among species. Phylogeny of eight bird species shows divergence times (denoted at the nodes). MMO: monk parakeet (*Myiopsitta monachus*); ACR: golden eagle (*Aquila chrysaetos*); PAU: double-crested cormorant (*Phalacrocorax auritus*). Vertical bars show the frequency of TE insertions during

the evolution of bird species. **e–g** Contents of DNA-hAT-Charlie, DNA (excluding hAT-Charlie), LTR in non-rearranged (NRA-chr), and rearranged chromosomes (RA-chr). Rearranged chromosomes in this context refer to the chromosomes with observed fusion events. $n = 6$ and 7 for NRA-chr and RA-chr, respectively. In order to avoid deviations caused by TE characteristics between micro- and macrochromosomes, only chromosomes with lengths longer than 20 Mb were used for statistics. The number on the horizontal line above each two boxes represents *p*-values (two-sided *t*-test). NS indicates no significant difference between the two groups of data. The line in the middle of box represents the median, the upper and lower boundaries of the box are the upper and lower quartiles, respectively. The boundaries of the upper and lower whiskers are the maxima and minima, respectively. Source data are provided as a Source Data file.

To validate the neo-Z/W chromosomes of crested ibis, we closely examined the fusion point between the ancient- Z/W and the added- Z/W (chromosome 22-derived parts), at ~85.5 Mb for the neo-Z and ~37.84 Mb for the neo-W, respectively (Supplementary Data 2 and 3). The Hi-C interaction maps strongly supported the assembled chromosome models (Fig. 2a). In addition, for both the neo-Z and W fusion points, a single contig spans the flanking regions, and the contigs are uniformly covered by long sequencing reads (Fig. 2b, Supplementary Fig. 5a–c). Furthermore, FISH experiments using specific probes for chicken chromosome 22 sequences showed that fluorescence signals were present on the pair of Z chromosomes of a crested ibis, which further validates that the neo-sex chromosomes are derived by a fusion event between the ancient-sex chromosomes and chromosome 22 (Fig. 2c).

The reduced female sequencing depth on the added-Z chromosome (chr22-derived) suggests that the added-Z and added-W have ceased recombination and have started differentiation following their addition to the ancient-sex chromosomes (Fig. 2a). We found that the added-Z can be divided into two distinct parts. The sequences proximate to the fusion point, ~3 Mb in size, are hemizygous and exhibit half sequencing depth in both the added-Z and the added-W, resembling the sex differentiation region (SDR) on ancient-sex chromosomes. The remaining sequences (~4 Mb), residing at the chromosomal end, are homozygous with autosomal-like sequencing depths (Fig. 1b, Fig. 2a). Therefore, we hypothesized that this ~4 Mb terminal region is a putative new pseudo autosomal region (new-PAR). Intriguingly, we identified a ~3 Mb inversion between the added-Z and the added-W that coincided with the boundary for the new-PAR (Supplementary Fig. 6). Both breakpoints for the inverted region are located within contigs, suggesting the correct assembly of this inversion (Supplementary Fig. 7). Thus, our analysis supports the role of a structural variation in the formation of a new SDR (Fig. 2d).

The ancient-PAR, located at the other tip of the sex chromosomes, (Fig. 2a, d) has a size of 3.24 Mb, similar to that of the new-PAR, but larger than those of most other birds[14,52,56]. The new-PAR and ancient-PAR both show high SNP density, in contrast to the hemizygous status of the SDRs (Supplementary Figs. 8a and 9).

### Evolution of a young stratum on the neo-sex chromosome
It is known that independent suppression of recombination between sex chromosomes has created three evolutionary strata (S0–S2) that are shared by all species of Neoaves. Most Neoaves lineages have independently evolved an additional stratum S3 with varying size[52,56]. Our analysis examining the divergence levels between the neo-Z and W chromosomes confirms the four strata (S0–S3) in the ancient-sex chromosomes, but also identified one new stratum (S4) in the added-sex chromosome (Fig. 2e).

Our analysis revealed that S4 exhibited ~92.5% sequence similarity between the added-W and added-Z, similar to that of S3 (91.9%) but higher than those of the older strata (S0 and S1:80.7%; S2:87.5%, Fig. 2e). The synonymous substitution rate (*dS*) of gametologous pairs

of S4 are lower than those of the older strata, but similar to those of the S3 stratum (Fig. 2f, Supplementary Data 4, Wilcoxon signed rank test), which indicates that the emergence time of S4 was close to that of S3. Moreover, we observed lower transposable densities (TEs) and higher gene densities in S4 relative to the other ancient strata (Supplementary Fig. 8b–d).

To investigate whether the neo-sex chromosomes are exclusive for the crested ibis, we carried out a synteny analysis between chicken and another ibis: plumbeous ibis (*Theristicus caerulescens*, TCA). The neo-Z chromosome fused by chr22 and ancient-Z was found in plumbeous ibis, representing a common sex chromosome-autosome fusion event in ibises (Supplementary Fig. 10). Besides, we conducted a phylogenetic analysis of the gametologous genes of the S4 strata, together with their orthologs from various bird species, including the plumbeous ibis and the black-faced spoonbill (*Platalea minor*, PMI), which belongs to the *Platalea* of Threskiornithidae. The results of this analysis show that Threskiornithidae genes are clustered by chromosomes rather than by species, suggesting that the added-sex chromosomes were likely formed in the common ancestor of the Threskiornithidae (Fig. 2g, Supplementary Fig. 11).

Together, the results of our analyses show that the formation of the neo-sex chromosomes is probably closely associated with the emergence of S3. It is speculated that the neo-sex chromosomes in Threskiornithidae emerged in an early common ancestor, before the first split within this family, but after they diverged from the Ardeidae family.

### Added-W chromosome evolved an ovary-biased gene category
To characterize the expression patterns of added-sex chromosome genes after they were translocated from an autosome to a sex chromosome pair, we compared the expression profiles of single-copy genes on the S4 strata (26 genes) and the new-PAR (94 genes, Supplementary Data 5), with their orthologous genes on chicken chromosome 22, across five different tissues. Our results showed significantly higher and broader expression of genes in both the added-Z and added-W (S4 and new-PAR) than their orthologs in chicken (GGA-chr22), and that this tendency was consistent across all tissues (Fig. 3a, b and Supplementary Fig. 12).

We further found that the pattern of high expression levels of added-sex chromosome genes still holds true when they were compared with crested ibis autosomal genes (Supplementary Fig. 13). This is in contrast to the expectation that sex-linked are typically expressed at a lower level compared with autosomal genes[59]. In fact, the ancient-sex chromosome-linked genes of the crested ibis do have a lower expression level (Supplementary Fig. 13), suggesting that the genes in the added-sex chromosome have not yet evolved a low-expression mode, possibly due to their relatively recent origin. Despite this, we still observed a substantial divergence of expression between the gametologous pairs, as we found almost no correlation ($r^2 = 0.00422$) between them across all five different tissues (Fig. 3c).

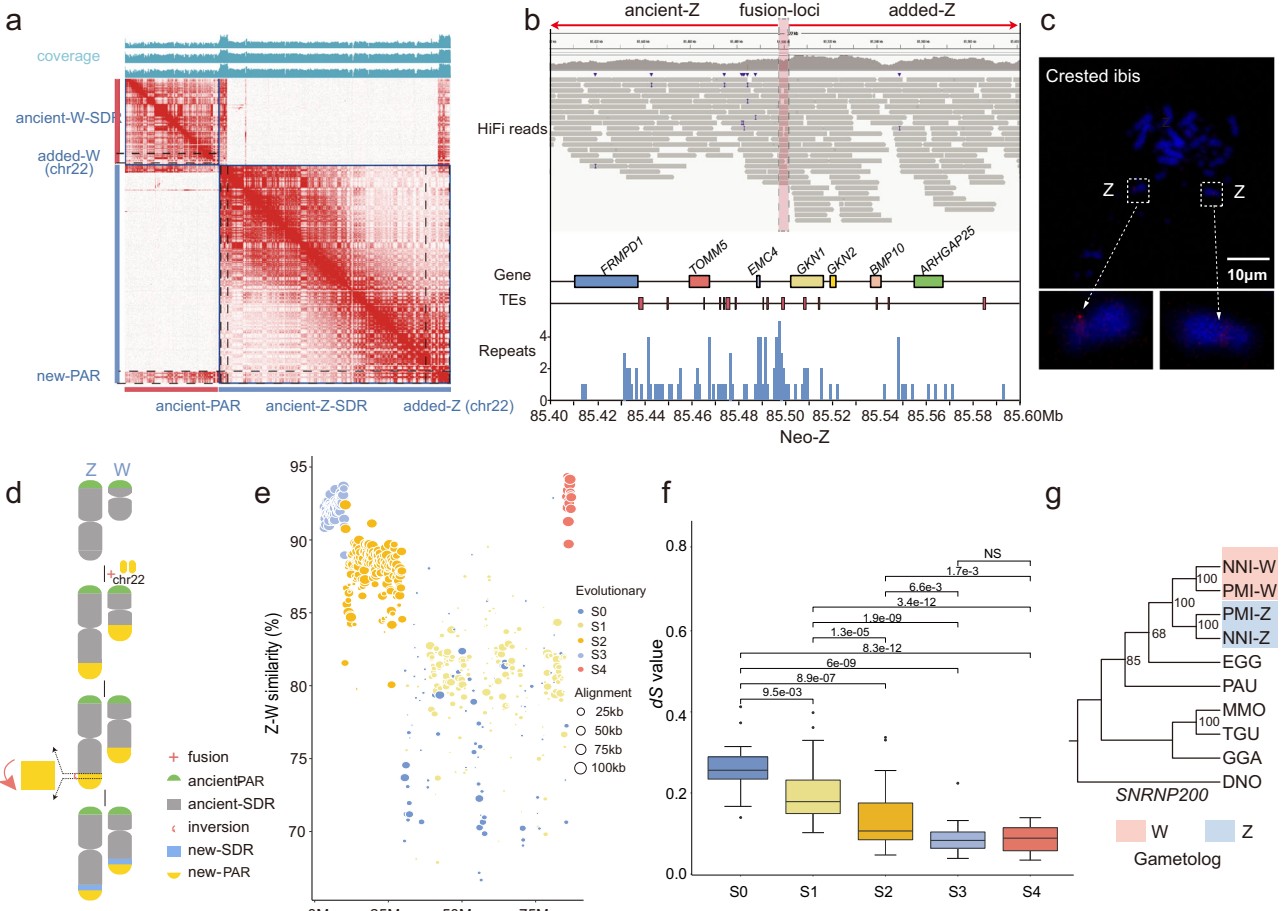

**Fig. 2 | The Evolutionary history of added-sex chromosomes of crested ibis.**
**a** Hi-C contact map of the neo-Z and neo-W of the crested ibis presenting territories homologous to chicken chr22. **b** IGV screenshot of the results of raw reads mapping to the neo-Z. The dashed line indicates the fusion site of the neo-Z, which is crossed by consecutive reads. The three panels display the distributions of genes, TEs, and repeats within 100 kb upstream and downstream of the fusion loci. **c** FISH images for the probes of chicken chr22 hybridized in crested ibis chrZ. The total length of probes is approximately 9 kb. The dashed boxes show the chrZ of crested ibis, and the arrows point towards the enlarged area within the dashed box. The Red dots are hybrid signals. The FISH experiments were repeated for twice with similar results. **d** Schematic diagram of the evolutionary process for the neo-sex chromosomes. After a part of chr22 underwent inversion, a new SDR (blue rectangle) formed, and the remaining part (yellow semicircle) formed a new-PAR. **e** Sequence divergence of the Z and W reveals the pattern of evolutionary strata. Different color shows each of the evolutionary strata. Size of circles is scaled to the length of sequence alignments. **f** $dS$ values for the Z−W gametologs of each stratum. The numbers of genes for S0, S1, S2, S3, and S4 are 14, 29, 79, 21, and 32, respectively. The number on the horizontal line above each two boxes represents $p$-values (two-sided Wilcoxon signed rank test). NS indicates no significant difference between two data groups. The line in the middle of the box represents the median, the upper and lower boundaries of the box represent the upper and lower quartiles, respectively. The boundaries of the upper and lower whiskers represent the maxima and minima, respectively. **g** Phylogeny of the Z−W gametologs for S4. EGG represents egret (*Egretta garzetta*, belongs to Ardeidae family). Threskiornithidae genes are clustered by sex chromosome rather than species, suggesting this autosomal-sexual chromosome fusion event happened after the divergence of Threskiornithidae and Ardeidae. Additional gene trees are given in Supplementary Fig. 11. Source data are provided as a Source Data file.

Although the Z-S4 and W-S4 genes tend to be broadly expressed, we found that the W-S4 has a higher proportion of ovary-specific genes than the Z-S4 (12.3% *vs.* 8.0%, Fig. 3d). In fact, this proportion is also higher than for other parts of the genome (expect for WS0–WS3, Fig. 3d). Among the ovarian-specific genes located in the W-S4, four single-copy genes (*GKN2W*, *BMP10W*, *PROM2W*, and *ADRA1AW*) were found to have shifted from a non-gonadal specificity to an ovarian specificity (Fig. 3e, Supplementary Fig. 14). For instance, in the chicken, the autosomal gene *BMP10* has a heart-specific expression and plays an important functional role in heart development[60]. The derived ovarian-specific expression of *BMP10W* in the crested ibis possibly implies a functional renovation.

## Slow degradation of the neo-W chromosome
The degree of degeneration of the W chromosome varies among bird lineages[52]. Unlike all published W chromosomes from neognathae birds[13,14,23,24,53], restricted degradation on its ancient-W is exhibited in

the crested ibis. First, synteny analyses showed that the ancient-W chromosome maintained highly conserved synteny blocks with the ancient-Z chromosome, which is different from that seen in birds where weak synteny exists between the Z and W chromosomes (Fig. 4a, Supplementary Fig. 15). Second, higher sequence similarities between the ancient-Z and W chromosomes were detected relative to Galloanserae (duck and chicken) or songbirds (Fig. 2e)[14,24]. Third, in terms of gene content, we found that 80% of the chromosome 22-homologous gene content were retained in the S4 of the added-W, with only four genes (4/75, 5.33%) pseudogenized (Supplementary Table 4). The ancient-W SDR retained 339 genes (Supplementary Data 6), including approximately 12.25% orthologous genes of the proto-sex-chromosome, almost five times more than that seen in the duck (2.55%)[24]. These genes consist of 206 single-copy genes and 133 multi-copy genes (Supplementary Table 5), which is different from the gene composition of other bird W chromosomes, which have very few multi-copy genes[23,61]. Additionally, all of these genes were observed to

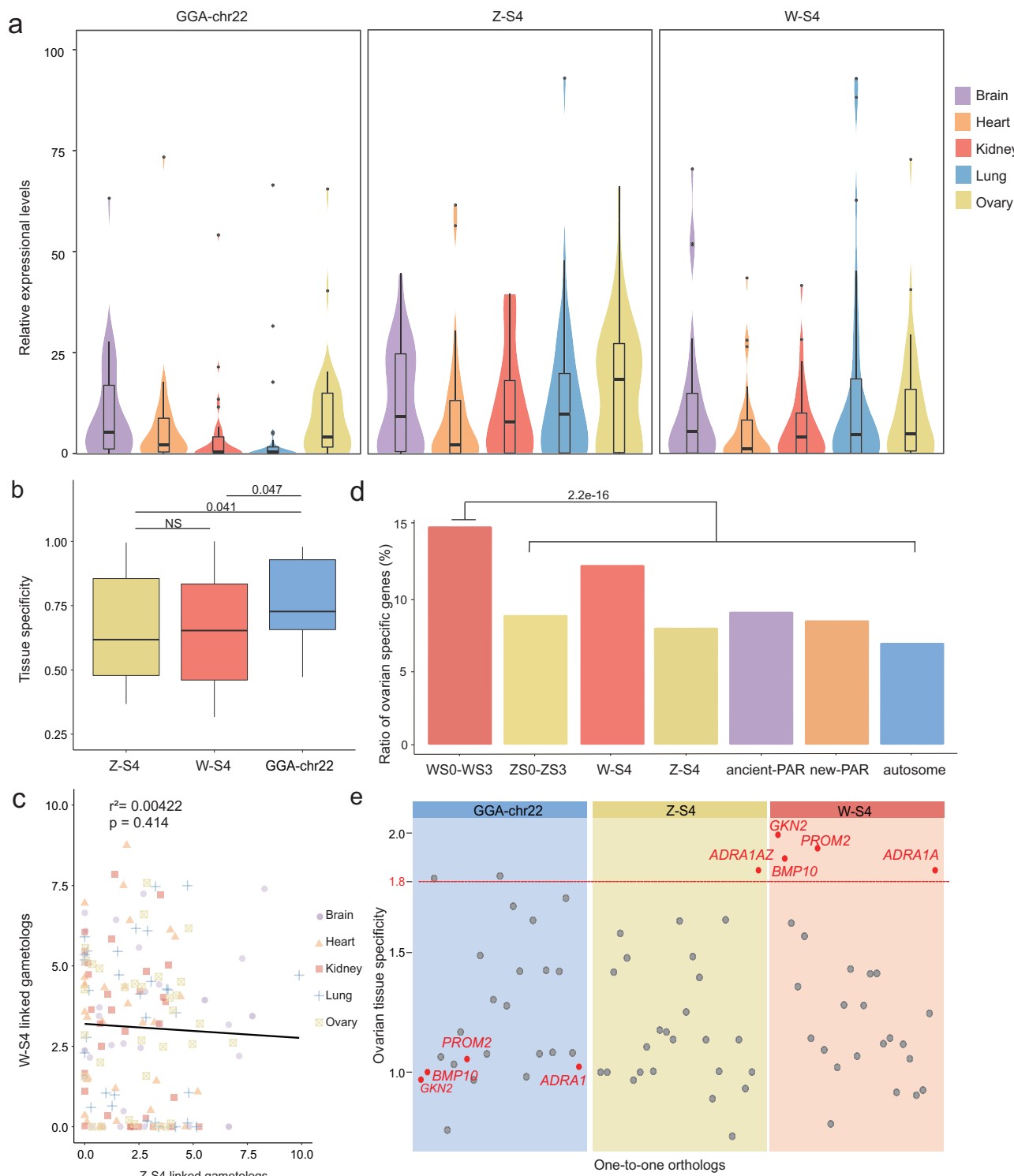

**Fig. 3 | Shifted gene expression patterns of the added-sex chromosomes of the crested ibis. a** Expression of Z-S4/W-S4 genes and their one-to-one orthologs from chromosome 22 of the chicken in different tissues. The number of orthologous genes is 28 for each group. **b** Tissue specificity (tested by Tau value) for Z-S4/W-S4 genes and their orthologs from chromosome 22 of the chicken. The number of orthologous genes is 25 for each group. The number on the horizontal line above every two boxes represents *p*-values (two-sided Wilcoxon signed rank test). The line in the middle of box represents the median of this set of data, the upper and lower boundaries of the box are the upper and lower quartiles, respectively. The boundaries of the upper and lower whiskers are the maxima and minima,

respectively. **c** The X- and Y-axis show the expression of Z-S4 and W-S4 linked gametologs in different tissues, respectively, which is measured by $\log_2(1 + FPKM)$. Linear regression was used with adjustments. The line represents the regression relationship between the expression of the Z and W genes. **d** Ratio of ovarian-specific genes in each region of the genome. Chi-Squared Test was used ($\chi^2 = 26.564$; $P = 2.2e\text{-}16$). **e** Ovarian tissue specificity of orthologs between Z-S4/W-S4 linked and chromosome 22 of the chicken. Genes above the red line show ovarian-specific expression patterns, with a Tau value higher than 0.8 and expressed mostly in the ovaries. Source data are provided as a Source Data file.

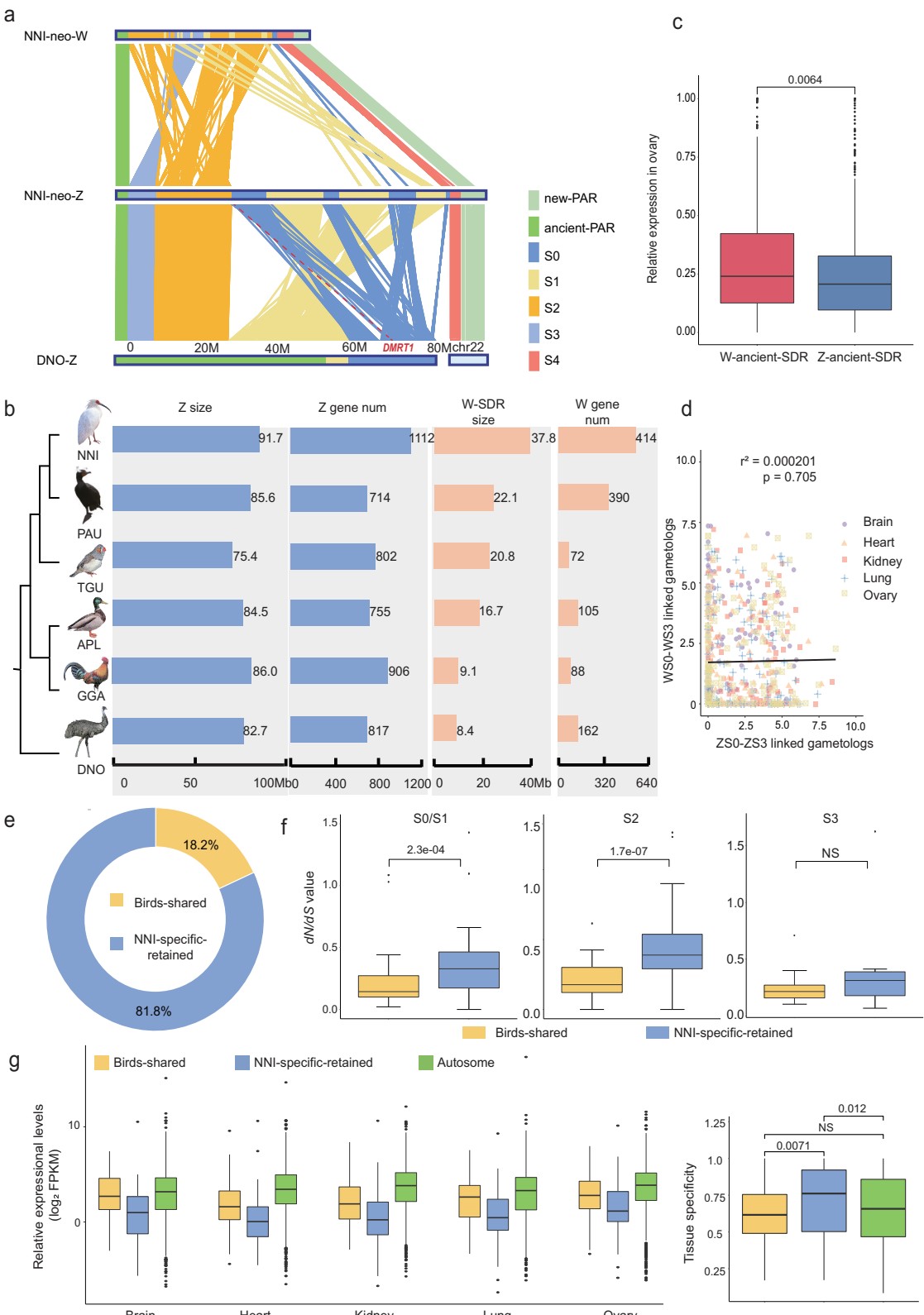

have Z-linked gametologs. Fourth, unlike the more than 40% pseudo-genized genes on the W-linked SDR of ducks[24], only 10.6% (36/339) of the genes on ancient-W in the crested ibis were pseudogenes, with 21 of these pseudogenes being multi-copy genes (Supplementary Tables 4 and 5). All of these findings suggested that the crested ibis has a low level of degeneration of the W chromosome.

We conducted a *dN/dS* analysis of the orthologous genes, and found that genes from the added-W and ancient-W both have a slow evolutionary rate, and that there was no significant difference in this ratio between the added-W and the ancient-W genes, which supports the conclusion that, similar to the ancient-W genes, genes on the added-W evolved at a slow evolutionary rate (Supplementary Fig. 16).

**Fig. 4 | Highly restricted degeneration and female-biased expression of the ancient-W in the crested ibis. a** Syntenic plot between the neo-Z and neo-W Chromosomes of the crested ibis and homologous chromosomes (Z and chromosome 22) of the emu. Each line represents one pair of aligned fragments, and each color corresponds to one evolutionary stratum. The emu strata are demarcated according to Liu et al.[21]. **b** Lengths and gene number of Z and W chromosomes across the studied bird species. **c** Expression in the ovary of Z- and W- linked genes on the ancient-SDR. $n = 294$ for genes of W-ancient-SDR, $n = 734$ for genes of Z-ancient-SDR. Two-sided Wilcoxon signed rank test was used. **d** The X- and Y-axis show the expression of Z- and W-linked gametologs in S0–S3, respectively. The solid line represents the regression relation between the expressions of the Z and W. Linear regression test with adjustments was used. **e** Proportion of the two groups of genes to all ovary-biased genes. **f** Comparison of the *dN/dS* values for the two groups of genes in the different strata. $n = 39$ and 65 for S0/S1 genes of Birds-

shared group and NNI-specific-retained group; $n = 36$ and 72 for S2 genes of Birds-shared group and NNI-specific-retained group; $n = 10$ and 18 for S3 genes of Birds-shared group and NNI-specific-retained group. Two-sided Wilcoxon signed rank test was used. **g** Left: Expression for two groups of genes on the ancient-SDR of the W and autosomes in different tissues. Right: Tissue specificity for two groups of genes on the ancient-SDR of the W and autosomes in different tissues. $n = 89$, 132, and 1273 for genes of Bird-shared group, NNI-specific-retained group, and Autosome group (chr1 was taken as example), respectively. The line in the middle of box represents the median of this set of data, the upper and lower boundaries of the box are the upper and lower quartiles, respectively. The boundaries of the upper and lower whiskers are the maxima and minima, respectively. Data statistical test was used two-sided Wilcoxon signed rank. Source data are provided as a Source Data file.

Thus, from the perspective of gene loss rate, and from gene evolutionary rate, these results suggest that, similar to the ancient-W, the added-W genes evolve at a slow rate.

We also performed a gene ontology analysis of the 339 genes present on the ancient-W SDR. These genes were enriched in cell proliferation and differentiation, among others (Supplementary Table 6), suggesting important functions in regulating cell growth and activity. This is similar to the mammalian Y-linked genes and the highly degenerated W chromosome genes found in birds[4].

To examine whether the low rate of W degeneration is shared by other core waterbirds, we analyzed the gene content of the W chromosome of the double-crested cormorant (*Phalacrocorax auratus*, PAU), which possesses similar ecological traits with the crested ibis, such as large body size, monogamous mating system, and sexual monomorphism[59]. A large number of genes (390 genes) were identified in the double-crested cormorant W chromosome (Fig. 4b).

Moreover, for two other sexually monomorphic and monogamous core waterbird species, the eastern white pelican (*Pelecanus onocrotalus*) and the pygmy cormorant (*Phalacrocorax pygmaeus*), previous cytological experiments showed large-sized W chromosomes[62]. These clues suggest a broader presence of restricted degradation of the W chromosome in large core waterbirds.

### Crested ibis-specific genes contribute to ovary-biased gene expression of the ancient-W chromosome

We further asked whether genes on the ancient-W chromosome of crested ibis also exhibit an ovary-biased expression pattern. To do so, we first filtered out ancient-W-linked genes with low FPKM values (<1) and performed an expression analysis on the remaining 221 genes. Our results revealed that the ovarian expression of the ancient-W is significantly higher than that of ancient-Z genes (31.8% vs. 24.5%, $P = 0.0064$, Wilcoxon signed rank) (Fig. 4c). Moreover, similar to the case of genes on S4, we detected no correlation between the expression of ancient-W and ancient-Z gametologs, in all of the examined tissues ($r^2 = 0.000201$, Fig. 4d). Furthermore, 14.9% of genes (33/221) on the SDR of the ancient-W chromosome were specifically expressed in ovaries, significantly higher than for ancient-Z-linked genes (Supplementary Fig. 17) and autosomal genes (Fig. 3d, $\chi^2 = 26.564$; $P = 2.2e\text{-}16$, Chi-Squared Test). This female-biased expression pattern of W-linked genes is distinct from our previous understanding of genes on the bird W chromosome[63].

To investigate the origin of the non-canonical expression pattern of the crested ibis ancient-W chromosome, we compared its gene content with that of other birds. We found only 89 ancient-W-linked genes that were shared by other birds (Birds-shared), while 132 genes were independently retained by the crested ibis (NNI-specific). Intriguingly, we found that ovary-biased genes were significantly enriched in the crested ibis-specific group ($\chi^2 = 7.870$; $= 0.006$, Chi-Squared Test), with 81.8% of the ovary-specific genes (27 out of 33,

Supplementary Data 7) clustered in the crested ibis-specific gene group, which account for 11.39% of all genes in this group, while only 18.2 % (6 out of 33) are in the birds-shared gene group, which represents only 5.82% of all genes in this group (Fig. 4e, Supplementary Table 7). These results suggest that the ovary-specific gene expression of the ancient-W chromosome is primarily due to genes that were specifically retained in the crested ibis.

Compared with the birds-shared group of genes, crested ibis-specific genes show a significantly faster evolutionary rate (Wilcoxon rank sum test, Fig. 4f), indicating that distinct selective pressures act on these two groups of genes. Additionally, we found that genes in the crested ibis-specific group had lower expression levels across all tissues, but a higher degree of tissue specificity (Wilcoxon rank sum test; Fig. 4g). This relationship between evolutionary rate and expression pattern is consistent with that observed in songbirds[14] and can be explained by the Misfolding Avoidance hypothesis[64].

## Discussion

By generating a high-quality de novo assembled female reference genome of the crested ibis, including the first chromosome-structured sex chromosomes, we were able to expand our knowledge and gain new insights into the evolution of sex chromosomes in core waterbirds species.

Dramatic variations in the evolutionary rate of sex chromosome differentiation are observed among the different avian lineages, ranging from the highly conserved homomorphic sex chromosomes in ratites[19,21] to the fully differentiated dimorphic sex chromosomes of the chicken[23]. The W chromosome of the crested ibis features a large SDR of approximately 37 Mb, making it one of the largest SDRs in known bird W chromosomes to date[65]. Furthermore, our observations of the large number of genes and high sequences similarity with the Z chromosome in the crested ibis' W chromosome, provide evidence for a highly restricted degeneration of the W chromosome, adding a new evolutionary pattern to bird sex chromosomes[24,66].

Our synteny analysis shed light on the ancient chromosomal fusion events that led to the decline in chromosome number in the crested ibis. In addition, we identified a case of microchromosomes fused with ancient-sex chromosomes, that was estimated to have occurred in the common ancestor of the Threskiornithidae family after divergence from Ardeidae, resulting in the formation of neo-sex chromosomes, a phenomenon that has not been previously reported in core waterbirds. Neo-sex chromosomes have been suggested to play a role in resolving sexual antagonism and may also affect speciation[53,66–69]. After autosomes transform to become part of a neo-sex chromosome, their constituent genes are expressed in an autosome-like high and broad pattern, which is opposite to the low and narrow expression of genes on the ancient-sex chromosomes[70]. This may suggest that genes in the added-sex chromosomes experienced strong constraints of purifying selection[71,72] that contributed to their survival on the neo-W chromosome[73,74].

Our study found that the ancient-W chromosome of the crested ibis experienced more restricted degradation compared to W chromosomes from other Neognathae species[57]. The level of degradation of heterogametic sex chromosomes is affected by various factors, such as mutation rate, sexual selection, species generation time, mating system, and effective population size[1,75]. The crested ibis has a large body size and a long generation time, which may have contributed to its relatively slow evolutionary rate. This is consistent with the low mutation rate found in bird species from aquatic orders[36], and may be one of the reasons for the observed low level of ancient-W degradation in the crested ibis.

Other efficient powers that contribute to the rapid degradation[76] of sex-limited chromosomes include mating system and sexual selection[71]. These have been suggested to have had major impacts on the disparity in mammalian Y chromosomes even between closely related species, such as human and chimpanzee[1,72,77]. In terms of chromosome evolution, this is reflected in the suppressed recombination between sex chromosomes[1]. The non-recombing regions of the W chromosome are severely degenerated due to increased sexual antagonism and the low efficiency of natural selection, resulting from Muller's ratchet and the Hill–Robertson effect[5,15,78]. In bird species with female heterogamety (ZW system), sexual dimorphism and male-targeted sexual selection[79,80] led to the excess accumulation of male fitness mutations on the Z chromosome, promoting a faster evolutionary rate for the Z chromosome, which is known as the 'fast-Z' effect[81,82]. This effect is observed more intensively in species with stronger sperm competition[83,84]. The accelerated evolution of the Z chromosome, in turn, may contribute to the suppression of recombination with the W chromosome[85]. However, in most core waterbirds, including the crested ibis, sexual monomorphism and monogamy may lead to the observed low degree of sexual antagonism. We hypothesize that the lower pressure of sexual selection, strict monogamy, and moderate sexual antagonism may be a potentially important factor affecting the low degree of degradation observed in the ancient-W chromosome of the crested ibis.

In mammals, the Y chromosome is characterized by the presence of male reproductive genes and may constitute a battleground for sexual selection[11,13]. However, unlike mammalian Y chromosomes[76], previous research has suggested that the W chromosome from birds is not associated with female-biased gene expression[13,14,53,63,86,87]. Surprisingly, in contrast to previous results from core land birds, the neo-W chromosome of the crested ibis appears to show female-biased expressions or ovarian-specific expressions from both ancient and new evolutionary strata. Furthermore, this female-biased expression pattern is driven by genes that have been specifically retained in the crested ibis but lost in other bird species. These retained genes occupied more than 70% of gene content of the crested ibis' W chromosome. This suggests a female-biased directional selection of the W chromosome in the crested ibis.

In female heterogametic systems, sexual selection acts on female fitness traits[88], leading to the preservation and up-regulation of genes on the W chromosome[89]. Recent research measured the gene expression levels of the W-linked genes in different breeds of domestic chicken and found that the genes associated with female fitness traits, such as increased female fecundity, showed a strong convergent pattern of up-regulation[90]. Conversely, genes linked to male fitness traits, such as aggression and plumage ornamentation, exhibited decreased expression levels, likely due to opposing selection pressures[90]. In the case of the crested ibis, which lacks strong sexual dimorphism and apparent sexual selection pressures on males[91], female-female competition may explain the observed bias towards W-linked gene expression in ovaries, to accumulate genetic fitness traits for female fertility, and as a consequence, may decelerate the degradation of the W chromosome.

In summary, our analyses of the crested ibis' sex chromosomes uncovered different evolutionary traits of the sex chromosomes of crested ibis compared to previously published results from land birds, highlighting the multidirectional evolution of bird sex chromosomes. There are over ten thousand extant bird species with diverse morphologies and behaviors. Sex chromosomes undoubtedly play an important role in the development and maintenance of these characters. While our understanding of the evolution of bird sex chromosomes still remains inadequate due to the limited number of species that have been investigated and the scarcity of high-quality and well-structured W chromosome data. Therefore, future work should focus on expanding our understanding by investigating more species and generating more comprehensive data on W chromosomes.

## Methods

### Sample collection and ethics statement

Tissues from a wild adult female crested ibis that had a failed rescue were used for HiFi, Hi-C sequencing, and RNA-seq. Brain, heart, kidney, lung, and ovary tissues from this individual were used for RNA-seq to quantify gene expression. In addition, blood samples were collected from three male and three female crested ibis individuals for resequencing to confirm the assembly of the sex chromosomes. Blood samples from a black-faced spoonbill individual were collected for resequencing. All samples of crested ibis were obtained from the Shaanxi Rare Wildlife Rescue Base, black-faced spoonbill samples were obtained from Guangzhou Wildlife Research Center. The study protocols received ethical approval from the Ethics Committee of the Guangzhou Wildlife Research Center (permit number: GZZOO2021C02) and Shaanxi Rare Wildlife Rescue Base (permit number: SRWRB202102). All experimental procedures were approved by the Animal Care and Use Committee of Shaanxi Normal University following the guidelines outlined in the Guide for the Care and Use of Laboratory Animals in China.

### Genome sequencing and assembly

A 15 Kb DNA SMRTbell library was constructed using a standard protocol for sequencing on the PacBio Sequel II platform with circular consensus sequencing (CCS)[92] to obtain long and accurate reads. HiFi reads were then de novo assembled using Hifiasm (v0.13) with default parameters[93] to generate an initial set of contigs. To improve assembly quality and generate a chromosome-level assembly, we constructed a Hi-C library by digesting cross-linked chromatin with the restriction enzyme MboI and sequencing it on the Illumina NovaSeq6000 platform. The resulting Hi-C raw reads were processed using the Juicer (v1.6)[94] pipeline to map them to the initial contigs. Next, we used 3D-DNA software (v201008)[95] to anchor the contigs onto draft chromosomes, which were then visualized using Juicebox (v1.11.08)[96]. Manual adjustment of the order and orientations of contigs along the chromosomes was performed to obtain a high-quality draft chromosome assembly. For resequencing work, we generated DNA libraries for each individual and sequenced them on the Illumina NovaSeq6000 platform with paired-end 150 bp read lengths and coverage of ~25x. These reads were used to calculate chromosomes sequence coverage and for SNPs calling. This comprehensive sequencing strategy allowed us to obtain a high-quality genome assembly for the downstream analyses.

To identify the chicken chr22 orthologs for the phylogenetic analysis, we additionally generated DNA libraries for a female black-faced spoonbill and generated ~50x Illumina short reads, these reads were assembled using SOAPdenovo version2.04[97] with parameters "-p 20 -R -K 51" to generate an initial set of contigs.

### Repetitive sequence annotation

To predict tandem repeats, we employed Tandem Repeats Finder (4.09)[98] and identified transposable elements (TEs) by searching the

protein database using RepeatProteinMask (RM-BLASTX) (Revision 1.23)[99]. We also constructed a de novo repeat library using Repeat-Modeler2 (2.0)[99] and performed TE analysis using RepeatMasker (open-4.0.7) with both the de novo library and the Repbase database. To further understand the evolutionary history of the TE sequences, we estimated their insertion times in emu, chicken, mallard, zebra finch, monk parakeet, golden eagle, double-crested cormorant, and crested ibis using a Kimura distance-based analysis[100] with the parseRM pipeline (https://github.com/4ureliek/Parsing-RepeatMasker-Outputs). To provide context, we employed the phylogeny of the birds from Jarvis et al[12].

### Protein-coding gene annotation

RNA-seq data was used for both genome annotation as well as for quantifying the expression of sex-linked genes. Assembled transcripts, along with the masked genome were used for training of de novo predicted coding genes using the Augustus software (version 2.5.5)[101]. Next, we conducted homology-based annotation by downloading proteins from the emu, chicken, mallard, monk parakeet, golden eagle, zebra finch, egret, and double-crested cormorant genomes from public databases (Supplementary Data 9) and selecting the longest transcript for each gene. We aligned this gene dataset to the genome assembly using blast+(tblastn) (2.10.1) with an e-value threshold of 1e-5 and predicted gene structures using GeneWise (v2.4.1)[102] with default parameters. We then aligned all of the assembled transcripts against the genome using BLAT[103] (version 34) with identity >90% and coverage >90%. We used PASA[104] (version 2.1.0) to filter overlapping sequences and link spliced alignments. Finally, EvidenceModeler v1.1.1[105] was used to integrate the results from the above analyses to obtain a final gene annotation.

### Chromosomal rearrangement and SNP calling

We employed a bioinformatics pipeline to perform pairwise whole-genome alignments with high accuracy and reliability. Specifically, we used the nucmer tool from MUMmer (4.0.0.beta2)[106] with the parameter "-b 400" to align the genomes. We then filtered the resulting alignments using delta-filt from the MUMmer package to retain only the one-to-one best hits. Finally, we formatted the alignments and visualized synteny using the MCscan JCVI utility pipeline v1.3.9[107]. To align the long reads to the reference assembly of the crested ibis, we utilized the Burrows–Wheeler Aligner (BWA, version 0.7.17-r1188)[108] with default parameters. We then used Samtools-1.9[109] and Picard Tools (Version 1.56, http://picard.sourceforge.ne) to sort bam files and filter duplicate reads. SNPs were called from all genomic alignments using the command HaplotypeCaller and GenotypeGVCFs from the Genome Analysis ToolKit package (version 3.8)[110]. We filtered out low-quality SNPs using the VariantFiltration command with the following criteria: "DP < 140 DP > 1260 QD < 2.0 FS > 60.0 MQ < 40.0 MQRankSum < -12.5 ReadPosRankSum < -8.0 SOR > 3.0".

### Validation of sequences added to sex chromosomes

After conducting the whole-genome alignment with the chicken, we identified fusion sites on the neo-sex chromosomes based on the alignment results (Supplementary Data 2 and 3). To confirm these fusion sites, we mapped raw reads from HiFi sequencing to the crested ibis genome using minimap2-2.17[111] with the parameters "-ax map-pb". We then used samtools-1.9[109] to generate bam files, which were manually checked using the Integrative Genomics Viewer (IGV) 2.11.3[112]. We used the same methods to validate the differentiated sites added to the sex chromosomes.

### Cell culture

A skin sample was taken from a male adult crested ibis individual. Immediately after disinfection, a $0.3 \times 0.6\ cm^2$ area of skin tissue was placed into a culture medium. The animal was placed back into the feeding net after bandaging. Methods for primary cell culture and cell proliferation, preparation of culture medium and growth medium required during the process of cultivating cells, as well as the concentration of pancreatin were all based on Wang et al[51].

### Chromosomes preparation

Fibroblast cells of crested ibis in the logarithmic growth phase were treated with colchicine at a final concentration of 0.2 µg/ml. After culturing for 3 h in the medium containing colchicine, cells were collected and then treated with a hypotonic solution (0.075 mol KCl) for 40 min. Cells were fixed using a fixing solution (methanol: acetic acid = 3:1). Finally, an appropriate amount of prepared cell suspension was dropped onto a microscope slide and air-dried.

### FISH experiments

A total length of ~9 kb conserved homologous sequences of chicken chr22 were chosen for probes synthesis. After designing primers, we amplified target fragments by PCR. We then followed the ToloPrep Gel Extraction and PCR Purification Kit protocol (Omega, #36113) to purify amplified DNA fragments. Specific probes were synthesized following the protocol of a Nick Translation Kit (Roche, #10976776001). The specific target chr22 sequences and primers were listed in Supplementary Data 8.

Chromosome slides were enzyme-digested and then prepared probes were hybridized to the chromosomes. Hybridization of labeled probes was detected using Anti-Digoxigenin-Rhodamine Fab fragments (Roche, #11207750910). Slides were subsequently incubated with DAPI and the sealed slides were visualized using confocal microscopy (Olympus FV3000, Tokyo, Japan).

### Evolutionary strata

Z and W chromosome sequences were masked and aligned by LASTZ (1.04)[113] with the relaxed parameters (--step=19 --hspthresh=2200 --inner=2000 --ydrop=3400 --gapped-thresh = 10,000), and alignment chains and nets were produced to join the syntenic blocks into longer alignments. Alignments with sequence identities higher than 96% or lower than 60%, or alignment lengths shorter than 65 bp were removed to eliminate misalignments or unmasked repeats[14]. We calculated sequence similarity between the Z and W chromosomes over 100-kb sliding windows along the Z chromosome. Windows based on sequence divergence levels were then plotted along the Z chromosome and we demarcated boundaries of evolutionary strata that displayed shifts in divergence level (excluding S0 and S1). Emu has a recent species-specific stratum (S1), while the oldest stratum (S0) is shared by all birds[52]. This allowed for the demarcation of S1 and S0 by synteny with the Z chromosome of the emu. The scripts used for these evolutionary strata analyses were downloaded from https://github.com/lurebgi/monkParakeet.

### Sex-linked gene analyses

To obtain one-to-one orthologous genes between Z and W chromosomes of the crested ibis, we carried out bi-directional blastn with parameters (-max_hsps 1 -max_target_seqs 1 -evalue 1e-5) on the annotated Z/W chromosome genes. The results with align ratio and sequence identity lower than 50% were omitted. According to the syntenic profiles upon alignments between the Z/W sequences, the identified Z/W one-to-one orthologous genes were assigned to the corresponding strata (S0-S4) for subsequent evolution rate analysis. We used MAFFT (v7.475)[114] with parameters (--globalpair --maxiterate 1000) to align Z/W orthologous genes. Finally, to calculate the $dN$ and $dS$ values of all orthologous genes, we ran the program megacc using MEGA-CC 11[115] software.

To identify one-to-one W-linked orthologous genes of different bird species, we used the software Orthofinder (version 2.5.4)[116] with default parameters. We then performed an all-versus-all bi-directional

blastn with the same parameters (see above) on these orthologous genes and retained the results with align ratio and sequence identity higher than 50%. The intersection of the two results from Orthofinder and blastn was used as the final dataset. Among these orthologous genes, we regrouped them into two categories: the birds-shared W gene group and the crested ibis (NNI)-W-specific-retained gene group. The first group consisted of conserved genes present on the W of the crested ibis and on the W of at least one other bird species; while the second group included W genes independently retained in the crested ibis but absent in all other birds.

To estimate the substitution rates of coding genes in the various species, we carried out multiple sequence alignments of orthologous genes. First, we used PRANK (v.170427)[117] to align sequences with default parameters. After filtering low-quality sites in the alignments, we estimated synonymous substitution rates ($dS$) and non-synonymous substitution rates ($dN$) using codeml from the PAML package (v.4.9e)[118]. Finally, we used RAxML (v. 7.0.4)[119] with the parameters (-T 3 -f a -x 12345 -# 100 -p 2 -m GTRGAMMA) to construct phylogenetic trees.

### Sex chromosome gene expression

Due to the endangered status of the crested ibis, we were only able to collect brain, heart, kidney, lung, and ovary tissues from a single female individual for RNA-seq (Supplementary Table 1). Unfortunately, we were unable to obtain samples from a male individual.

The five tissues collected for RNA-seq from the one female crested ibis were immediately frozen in liquid nitrogen. In addition, we downloaded raw RNA-seq reads from chicken tissues, including brain, heart, kidney, lung, and ovary, from GEO (Supplementary Data 9). We used HISAT2 (2.2.0)[120] to map the RNA reads with default parameters. After sorting the alignments, we used the RSEM pipeline (v.1.3.0)[121] to quantify gene expression levels. This pipeline employed STAR (v.2.5.30)[122] with default parameters to align raw reads to the transcriptomes constructed based on gene annotations. The expectation–maximization (EM) algorithm was then used to estimate abundance with RSEM. To normalize gene expression levels for each tissue, we calculated FPKM[123] (transcripts per million) and estimated the tissue specificity of the gene expression by calculating tau[124].

### Reporting summary

Further information on research design is available in the Nature Portfolio Reporting Summary linked to this article.

## Data availability

The assembled chromosome-leveled reference genome of the crested ibis has been deposited in the GenBank with the assembly accession ID of GCA_035839065.1 [https://www.ncbi.nlm.nih.gov/assembly/GCA_035839065.1/?shouldredirect=false]. The project accession number of this work that can be accessed in NCBI database is PRJNA974878 [https://www.ncbi.nlm.nih.gov/sra/?term=PRJNA974878], and all raw sequencing data generated by the research has been deposited in NCBI with the SRA accessions from SRX20466107 to SRX20466118, and SRX22371138. KEGG database [https://www.genome.jp/kegg/], Gene Ontology database [https://www.geneontology.org/] and Repbase database (Release 16.10) [https://www.girinst.org/repbase/index.html] were used in this work. A full list of accession IDs for public data is available in the Supplementary Data 9. Source data are provided as a Source Data file. Source data are provided with this paper.

## Code availability

All the data were analyzed by publicly available software and packages in this study. All involved software and packages in this study are described in the Methods section. Scripts and pipelines used for sex chromosomes analysis were downloaded from https://github.com/lurebgi/BOPsexChr.

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

## Acknowledgements

We would like to express our gratitude to Crested Ibis Artificial Breeding Center of Shaanxi Hanzhong Crested Ibis National Nature Reserve for the sample collection. We thank the B10K project, VGP Project, and Erich Jarvis at the Rockefeller University, M. Thomas P Gilbert at the University of Copenhagen, and Gary Graves at the Smithsonian for early access to the plumbeous ibis' genome. Unpublished Genome assembly and sequencing data of *Phalacrocorax auritus* are used with permission from the DNA Zoo Consortium (dnazoo.org). The photo of the crested ibis was taken by Caomu Guzi (https://www.birdnet.cn/portal.php). This work is supported by the National Natural Science Foundation of China (32270541 and 31872245 to X.Y., 31970391 to G.L. and 32000383 to Y.R.), the Natural Science Basic Research Program of Shaanxi (2020JM-280 to G.L.), the Fundamental Research Funds for the Central Universities (GK201902008 to G.L.), and Doctoral exploration project of Shaanxi Normal University (2021TS063 to Lulu Xu).

## Author contributions

G.L., X.Y. and Luohao Xu. conceived and designed the study. Lulu Xu. assembled the genome. Y.R. conducted the genome annotation. R.D. and W.C. conducted the sample preparation for genome analysis. Lulu Xu, Y.R., P.Y., J.Y., X.X. and D.M. carried out the data analyses. J. Wu., T.C., Z.F. and J.L. contributed to the cytogenetic works. C.H. and Lulu Xu performed the transcriptome analysis. Lulu Xu and T.Z. prepared the graphs. G.L. and Lulu Xu wrote the primary manuscript. G.L., Luohao Xu, X.Y., D.M.I., Y.Y. and J. Wang revised the manuscript. G.L., X.Y. and Luohao Xu supervised this study.

## Competing interests

The authors declare no competing interests.
