## [Peer Review File · Nature Communications]

REVIEWER COMMENTS

Reviewer #1 (Remarks to the Author):

The paper by Xu et al provides a new assembly and analysis of a genome assembly for the crested ibis. The authors find that the sex chromosomes in this species are interesting because the W is substantially larger than found in other birds and there appears to be a fusion of the Z and W with a microchromosome which results in neo-sex chromosomes in this species. Many of the analyses that the authors attempt are inline with what other neo-sex chromosome papers do (i.e., looking for synteny, evolutionary strata, TE changes, and gene expression changes) and in that regard it is good because it is somewhat consistent with others' work in other systems (e.g., *Drosophila*, nematodes, fish), but it also doesn't break any new ground other than this being a bird with ZW chromosomes. I found the results interesting, but unfortunately, I found reading the manuscript to be very challenging in large part because the English and writing needs to be improved. I would suggest that the authors work with someone to improve the English throughout the manuscript which will help with clarity and telling the story. There are numerous areas where what is being said is not accurate or is confusing and I assume this arose due to some translation issues. Below are some more major and minor issues that the authors should consider addressing.

Major

Line 123-126. The authors really need to include a plot showing Illumina sequencing coverage from a male and a female over the Z and W and compare that to the autosomes (or one reference autosome). The gene numbers and all downstream analyses and interesting results all are contingent on the sex chromosomes being assembled and annotated correctly and the more the authors can do to make it clear that they are, the more convincing the results. Figure 1a makes it look like the W and Z were some of the most fragmented chromosomes in the assembly.

Line 177. I think the normal convention is to call the old sex chromosome part the 'ancestral' XY(ZW) and call the newly fused chromosome the 'neo' XY(ZW). I think this might be easier for readers and you wouldn't need to add an unnecessary acronym (i.e., ASCRs). This also comes up again later in a different form (e.g., line 212) where the authors seem to be referring to the entire chromosome as 'neo-Z and neo-W chromosome' and then say there are regions that are ancient as well as ASCRs. I think the authors should reference some other papers regarding neo-sex chromosomes and borrow their terminology so that readers can follow the paper more easily.

The synteny analysis seems to leave out a lot of birds that have chromosome-level assemblies and that would provide an interesting contrast. Why? Looking at the parrot neo-sex chromosome paper (Huang et al 2022), Chr 22 doesn't even appear to be present in some species. Where does it go? What genes are on it that seem to not be essential in these other species?

Figure 2a seems to show that the two PARs were assembled on the Z but not on the W. Correct? I think being more explicit about this would help, there is a lot to digest in lines 180-192. Doesn't PacBio hifi result in haplotype level assemblies? Were the authors unable to assemble the W PAR region?

Is the one New-PAR really a PAR as it is typically defined, or is it just the part of chr22 that still recombines like an autosome? I imagine the scenario here is that in the past, chr22 fused to the Z and W. Initially, they would have just recombined like they would have when they weren't fused to the Z and W. Over time, an inversion occurred that limited recombination between parts of chr22, and at that point, those segments of chr22 can start to differentiate from each other. Right? I think this should be made clearer either through a figure or in the text.

Minor.

Lines 14-37. The abstract needs a significant overhaul. There are multiple places where the writing (English) needs to be improved for clarity.

Lines 40-47. I think it is important that the authors make it very clear that they are just talking about birds and mammals here and that what happens to sex chromosomes can be very different outside of these two groups.

Line 153 and Figure 1e. What is 'rearranged' in this context? Fused?

Figure 2b. I would consider reworking this figure so that it is clearer how much flanking sequence there is on either side of the putative fusion point and also what is in that picture. Are there any genes? Is this region all repetitive elements?

Line 182-184 and elsewhere. I would consider changing 'back' and 'front' to something else. This is confusing.

Figure 2c. I would consider adding a chr22 boundary to the figure to help reinforce that the inversion (new-Z/W-SDR) is only a part of the fused chromosome.

The link between sexual selection/sexual antagonism and sex chromosome evolution is not well referenced and also still up for debate (see Ruzicka F, Connallon T. 2020). I think the authors should consider this in the intro and discussion and adjust and cite accordingly.

Fig 2b. Do you know where the centromere is located on the Z and W?

Figure 4a would benefit from having all the birds they are discussing in the text be shown in the synteny comparison. On a related note, it is confusing how lines between the Neo-z and DNO-Z don't connect segments of the same color. Why are S1 stratum shown with yellow lines connected to green parts of DNO-Z?

Figure 4b. Add numbers over the bars. You can see broad patterns with the way things are plotted but the numbers would be more informative.

Reviewer #2 (Remarks to the Author):

Recent advances in avian genomics, particularly studies non-model organisms, consistently unveils novel insights into genome organization and evolution. This also applies to the sex chromosomes of birds, which is a subject of manuscript of Xu et al. The evolutionary trajectory of avian sex chromosomes seems to be much more confused and multidirectional than one might think. The present study addresses an important gap in the understanding of sex chromosome evolution in core waterbirds.

Using a comprehensive analysis with help of 3rd generation Hi-Fi sequencing methods, the authors revealed two characteristic features of the crested ibis' sex chromosomes: 1) their fusion with the autosome forming neo-sex chromosomes; 2) slow evolutionary rate of W chromosome. The evidence of both features are convincing and the methodology appropriately described. However, the obtained results should be discussed and interpreted more carefully.

Both features are in line with the theory of sexual selection and support the hypothesis of slower sex chromosome evolution in species with low sexual antagonism. However, this study certainly does not give "a comprehensive understanding of the associations between the sex chromosome evolution and the effects from sex selection or antagonisms" (lines 74-75). Authors highlight this potential connection consistently in the manuscript, exemplified by its placement at the outset of the Abstract. However, their data in no way actually test this association since only one species is analyzed.

Moreover, authors do not place these implications to the broader evolutionary context. Indirect data suggest that birds from other taxa may have restrict degradation of W (lines 297-304). Do they possess the same ecological features and patterns of sexual selection? Another feature that is not discussed in this context is the inversion between the neo Z- and W-linked ASCRs, which thus demonstrate pattern typical for W chromosome degradation through recombination suppression. Is it consistent with the slow evolutionary rate of ancient-W?

As for the formation of neo-sex chromosome, the fact that long reads overlap the fusion region is

rather reliable proof for the fusion between sex chromosomes and autosomes. However cytological evidence, thus, FISH with chromosome- or region- specific probe, would be still of much value. Perhaps it is worth trying to do this on a related species that does not have endangered status. This would solve two problems – prove the fusion and show that this event is not exclusive for the crested ibis. I would also like to see at least a brief discussion of the mechanisms leading to this unusual, though not unique, fusion.

Minor comments:

Line 99: Only from the title of this section, one can learn that this assembly of the crested ibis chromosomes is not the first. I think previous attempts should be referred.

Line 106: I would classify chromosomes 1-9 as macrochromosomes, and the rest as micro-. The Fig. 1a of the manuscript and the karyotype from Fig. 5 in Wang et al. (2012) clearly show the size gap between chromosomes 9 and 10, which is much larger than between 11 and 12. Moreover, this will be more consistent with the karyotypes of other birds, given the fusions in macrochromosomes in crested ibis.

Line 107-110: First, it is somewhat disappointing that the Hi-Fi reads assembly still does not reach enough resolution to distinguish the smallest microchromosomes. Second, I find this somewhat unfair to claim that "very few non-repetitive genomic sequences are missing from the chromosome assembly" since 6 more microchromosomes did not assemble (which should approximately comprise at least 20-40 Mb based on cytogenetic data). Its importance is probably underestimated given that "micro-chromosomes contained a significantly higher gene density and GC content, but less TEs" especially in light of recent discoveries like Li et al [doi.org/10.1093/molbev/msac066].

Line 146-148: I am just wondering in what extent such a comparison of data from comprehensive Hi-Fi assembly with the data obtained by second generation sequencing technology (for which repeats possess a known difficulty) is plausible.

Line 154: Please provide a quantitative characteristic of what you meant by the "chromosomal terminal" (at least in Supplementary).

Line 196: What do you mean claiming that the ancient-PAR "had lengthened by 3.24Mb"? Compared to what?

Line 234: I would added an estimated time of neo-sex chromosome formation here.

Line 281-287: I think the phenomenon of higher proportion of multi-copy genes in ancient-W deserves more attention. Are these copies functional or pseudogenes? Do they have active homologues on Z?

In Supplementary fig 12 the letters of subfigures do not correspond figure capture.

The text demands substantial grammar and style revision. A number of sentences are hard to follow, contain redundancy and/or typos (e.g. line 517, 531).

Thus, the manuscript makes a significant contribution to the field of avian genomics. However, it should be revised to include important details and discuss the results more carefully.

RESPONSE TO REVIEWERS' COMMENTS

Reviewer #1 (Remarks to the Author):

The paper by Xu et al provides a new assembly and analysis of a genome assembly for the crested ibis. The authors find that the sex chromosomes in this species are interesting because the W is substantially larger than found in other birds and there appears to be a fusion of the Z and W with a microchromosome which results in neo-sex chromosomes in this species. Many of the analyses that the authors attempt are inline with what other neo-sex chromosome papers do (i.e., looking for synteny, evolutionary strata, TE changes, and gene expression changes) and in that regard it is good because it is somewhat consistent with others' work in other systems (e.g., *Drosophila*, nematodes, fish), but it also doesn't break any new ground other than this being a bird with ZW chromosomes. I found the results interesting, but unfortunately, I found reading the manuscript to be very challenging in large part because the English and writing needs to be improved. I would suggest that the authors work with someone to improve the English throughout the manuscript which will help with clarity and telling the story. There are numerous areas where what is being said is not accurate or is confusing and I assume this arose due to some translation issues. Below are some more major and minor issues that the authors should consider addressing.

Major

Line 123-126. The authors really need to include a plot showing Illumina sequencing coverage from a male and a female over the Z and W and compare that to the autosomes (or one reference autosome). The gene numbers and all downstream analyses and interesting results all are contingent on the sex chromosomes being assembled and annotated correctly and the more the authors can do to make it clear that they are, the more convincing the results. Figure 1a makes it look like the W and Z were some of the most fragmented chromosomes in the assembly.

Line 177. I think the normal convention is to call the old sex chromosome part the 'ancestral' XY(ZW) and call the newly fused chromosome the 'neo' XY(ZW). I think this might be easier for readers and you wouldn't need to add an unnecessary acronym (i.e., ASCRs). This also comes up again later in a different form (e.g., line 212) where the authors seem to be referring to the entire chromosome as 'neo-Z and neo-W chromosome' and then say there are regions that are ancient as well as ASCRs. I think the authors should reference some other papers regarding neo-sex chromosomes and borrow their terminology so that readers can follow the paper more easily.

The synteny analysis seems to leave out a lot of birds that have chromosome-level assemblies and that would provide an interesting contrast. Why? Looking at the parrot neo-sex chromosome paper (Huang et al 2022), Chr 22 doesn't even appear to be present in some species. Where does it go? What genes are on it that seem to not be essential in these other species?

Figure 2a seems to show that the two PARs were assembled on the Z but not on the W. Correct? I think being more explicit about this would help, there is a lot to digest in lines 180-192. Doesn't PacBio hifi result in haplotype level assemblies? Were the authors unable to assemble the W PAR region?

Is the one New-PAR really a PAR as it is typically defined, or is it just the part of chr22 that still recombines like an autosome? I imagine the scenario here is that in the past, chr22 fused to the Z and W. Initially, they would have just recombined like they would have when they weren't fused to the Z and W. Over time, an inversion occurred that limited recombination between parts of chr22, and at that point, those segments of chr22 can start to differentiate from each other. Right? I think this should be made clearer either through a figure or in the text.

Minor.

Lines 14-37. The abstract needs a significant overhaul. There are multiple places where the writing (English) needs to be improved for clarity.

Lines 40-47. I think it is important that the authors make it very clear that they are just talking about birds and mammals here and that what happens to sex chromosomes can be very different outside of these to groups.

Line 153 and Figure 1e. What is 'rearranged' in this context? Fused?

Figure 2b. I would consider reworking this figure so that it is clearer how much flanking sequence there is on either side of the putative fusion point and also what is in that picture. Are there any genes? Is this region all repetitive elements?

Line 182-184 and elsewhere. I would consider changing 'back' and 'front' to something else. This is confusing.

Figure 2c. I would consider adding a chr22 boundary to the figure to help reinforce that the inversion (new-Z/W-SDR) is only a part of the fused chromosome.

The link between sexual selection/sexual antagonism and sex chromosome evolution is not well referenced and also still up for debate (see Ruzicka F, Connallon T. 2020). I think the authors should consider this in the intro and discussion and adjust and cite accordingly.

Fig 2b. Do you know where the centromere is located on the Z and W?

Figure 4a would benefit from having all the birds they are discussing in the text be shown in the synteny comparison. On a related note, it is confusing how lines between the Neo-z and DNO-Z don't connect segments of the same color. Why are S1 stratum shown with yellow lines connected to green parts of DNO-Z?

Figure 4b. Add numbers over the bars. You can see broad patterns with the way things are plotted but the numbers would be more informative.

Reviewer #2 (Remarks to the Author):

Recent advances in avian genomics, particularly studies non-model organisms, consistently unveils novel insights into genome organization and evolution. This also applies to the sex chromosomes of birds, which is a subject of manuscript of Xu et al. The evolutionary trajectory of avian sex chromosomes seems to be much more confused and multidirectional than one might think. The present study addresses an important gap in the understanding of sex chromosome evolution in core waterbirds. Using a comprehensive analysis with help of 3rd generation Hi-Fi sequencing methods, the authors revealed two characteristic features of the crested ibis' sex chromosomes: 1) their fusion with the autosome forming neo-sex chromosomes; 2) slow evolutionary rate of W chromosome. The evidence of both features are convincing and the methodology appropriately described. However, the obtained results should be discussed and interpreted more carefully.

Both features are in line with the theory of sexual selection and support the hypothesis of slower sex chromosome evolution in species with low sexual antagonism. However, this study certainly does not give “a comprehensive understanding of the associations between the sex chromosome evolution and the effects from sex selection or antagonisms” (lines 74-75). Authors highlight this potential connection consistently in the manuscript, exemplified by its placement at the outset of the Abstract. However, their data in no way actually test this association since only one species is analyzed.

Moreover, authors do not place these implications to the broader evolutionary context. Indirect data suggest that birds from other taxa may have restrict degradation of W (lines 297-304). Do they possess the same ecological features and patterns of sexual selection? Another feature that is not discussed in this context is the inversion between the neo Z- and W-linked ASCRs, which thus demonstrate pattern typical for W chromosome degradation through recombination suppression. Is it consistent with the slow evolutionary rate of ancient-W?

As for the formation of neo-sex chromosome, the fact that long reads overlap the fusion region is rather reliable proof for the fusion between sex chromosomes and autosomes. However cytological evidence, thus, FISH with chromosome- or region- specific probe, would be still of much value. Perhaps it is worth trying to do this on a related species that does not have endangered status. This would solve two problems – prove the fusion and show that this event is not exclusive for the crested ibis. I would also like to see at least a brief discussion of the mechanisms leading to this unusual, though not unique, fusion.

Minor comments:

Line 99: Only from the title of this section, one can learn that this assembly of the crested ibis chromosomes is not the first. I think previous attempts should be referred.

Line 106: I would classify chromosomes 1-9 as macrochromosomes, and the rest as micro-. The Fig. 1a of the manuscript and the karyotype from Fig. 5 in Wang et al. (2012) clearly show the size gap between chromosomes 9 and 10, which is much larger than between 11 and 12. Moreover, this will be more consistent with the karyotypes of other birds, given the fusions in macrochromosomes in crested ibis.

Line 107-110: First, it is somewhat disappointing that the Hi-Fi reads assembly still does not reach enough resolution to distinguish the smallest microchromosomes. Second, I find this somewhat unfair to claim that "very few non-repetitive genomic sequences are missing from the chromosome assembly" since 6 more microchromosomes did not assemble (which should approximately comprise at least 20-40 Mb based on cytogenetic data). Its importance is probably underestimated given that "micro-chromosomes contained a significantly higher gene density and GC content, but less TEs" especially in light of recent discoveries like Li et al [doi.org/10.1093/molbev/msac066]).

Line 146-148: I am just wondering in what extent such a comparison of data from comprehensive Hi-Fi assembly with the data obtained by second generation sequencing technology (for which repeats possess a known difficulty) is plausible.

Line 154: Please provide a quantitative characteristic of what you meant by the "chromosomal terminal" (at least in Supplementary).

Line 196: What do you mean claiming that the ancient-PAR "had lengthened by 3.24Mb"? Compared to what?

Line 234: I would add an estimated time of neo-sex chromosome formation here.

Line 281-287: I think the phenomenon of higher proportion of multi-copy genes in ancient-W deserves more attention. Are these copies functional or pseudogenes? Do they have active homologues on Z?

In Supplementary fig 12 the letters of subfigures do not correspond figure capture.

The text demands substantial grammar and style revision. A number of sentences are hard to follow, contain redundancy and/or typos (e.g. line 517, 531).

Thus, the manuscript makes a significant contribution to the field of avian genomics. However, it should be revised to include important details and discuss the results more carefully.

Response to reviewers:

Reviewer #1:

The paper by Xu et al provides a new assembly and analysis of a genome assembly for the crested ibis. The authors find that the sex chromosomes in this species are interesting because the W is substantially larger than found in other birds and there appears to be a fusion of the Z and W with a micro-chromosome which results in neo-sex chromosomes in this species. Many of the analyses that the authors attempt are inline with what other neo-sex chromosome papers do (i.e., looking for synteny, evolutionary strata, TE changes, and gene expression changes) and in that regard it is good because it is somewhat consistent with others' work in other systems (e.g., *Drosophila*, nematodes, fish), but it also doesn't break any new ground other than this being a bird with ZW chromosomes. I found the results interesting, but unfortunately, I found reading the manuscript to be very challenging in large part because the English and writing needs to be improved. I would suggest that the authors work with someone to improve the English throughout the manuscript which will help with clarity and telling the story. There are numerous areas where what is being said is not accurate or is confusing and I assume this arose due to some translation issues. Below are some more major and minor issues that the authors should consider addressing.

R: Thanks for this comment and we appreciate the positive remarks. We apologize for the language defects, and in the revised version, we have extensively polished the English writing throughout the entire manuscript. We hope the revised version is clearer to the readers.

Major issues:

Q: Line 123-126. The authors really need to include a plot showing Illumina sequencing coverage from a male and a female over the Z and W and compare that to the autosomes (or one reference autosome).

R: Thanks for pointing this out. In the revised manuscript, we added a new plot (Fig. 1b) to show sequencing coverage of the Z and W chromosomes and an autosome (Chromosome 5, which is of similar size to the Z chromosome) using whole genome resequencing data from male and female individuals. This mapping result shows that the Z chromosome uniformly exhibits a 2-fold elevation of sequencing coverage in male compared to the female, while the sequencing coverage of the W chromosome in the female crested ibis exhibits a read depth on average half of that of autosomes. These results are in line with expectations for differentiated ZW sex chromosomes. We have rewritten these sentences in the revised manuscript:

“By contrasting male and female sequencing coverage, we identified the Z and W chromosomes, where in females the Z and W show half the autosome sequencing coverage and in males the W has sparse male coverage (Fig. 1b).”

“Chromosome Z uniformly exhibits a 2-fold elevation of sequencing coverage in males relative to females (Fig. 1b), in agreement with the expectation for a differentiated Z chromosome.”

Q: The gene numbers and all downstream analyses and interesting results all are contingent on the sex chromosomes being assembled and annotated correctly and the more the authors can do to make it clear that they are, the more convincing the results. Figure 1a makes it look like the W and Z were some of the most fragmented chromosomes in the assembly.

R: Many thanks for the helpful suggestions. We are fully aware of the importance of having a well assembled and annotated Z and W sex chromosome sequences. In the revised manuscript, we provide more evaluation of the quality of the Z and W sex chromosome assemblies. First, as suggested by the reviewer above, we added a figure showing the male and female coverage on the Z and W chromosomes. Second, as pointed out by the reviewer, we removed Figure 1a which could be misleading.

Q: Line 177. I think the normal convention is to call the old sex chromosome part the ‘ancestral’ XY(ZW) and call the newly fused chromosome the ‘neo’ XY(ZW). I think this might be easier for readers and you wouldn’t need to add an unnecessary acronym (i.e., ASCRs). This also comes up again later in a different form (e.g., line 212) where the authors seem to be referring to the entire chromosome as ‘neo-Z and neo-W chromosome’ and then say there are regions that are ancient as well as ASCRs. I think the authors should reference some other papers regarding neo-sex chromosomes and borrow their terminology so that readers can follow the paper more easily.

R: Thanks for the valuable suggestion. We fully agree that involving this acronym can lead to naming complexity and confusion, which may obstruct understanding by the readers. Following the reviewer’s suggestion, in the revised manuscript, we have uniformly redefined the ancient regions of the Z/W chromosomes as “ancient-Z/W” and the recently fused regions of the Z/W chromosomes as “added-Z/W” throughout the whole manuscript. Correspondingly, the entire sex chromosomes (including both the ancient and added regions) was named as the neo-sex chromosomes (neo-Z and neo-W), these terminologies were borrowed from blue-faced honeyeater neo-sex chromosome paper¹.

Q: The synteny analysis seems to leave out a lot of birds that have chromosome-level assemblies and that would provide an interesting contrast. Why?

R: We appreciate this comment. The karyotypes of birds are generally stable, while in some lineages, such as parrots, species or lineage-specific intense karyotypical changes have been observed, such as the karyotype pattern of the parrot being derived by lineage-specific rearrangements across multiple chromosomes that were not present in early bird ancestors².

In this work, the major purpose of involving the synteny analysis was to exhibit the major karyotype changes in the crested ibis compared to the conserved bird karyotype, which represents the ancestral status. Further, a figure that shows a synteny analysis involving a large number of complex lines displaying species and lineage-specific events in other birds may reduce ability to focus on crested ibis-specific events.

At the same time, we are aware that sampling of more representative bird orders

will be helpful to expand our views on a broad comparison of the karyotype patterns between crested ibis and other birds. In our revised manuscript, we add two more bird species, bustard³ (belonging to Gruiformes) and California condor⁴ (belonging to Accipitriformes) into the synteny analysis to draw a more extensive syntenic graph. The new results of our synteny comparison and the new figure (Fig. 1c) are included in the revised manuscript.

Q: Looking at the parrot neo-sex chromosome paper (Huang et al 2022), Chr 22 doesn't even appear to be present in some species. Where does it go? What genes are on it that seem to not be essential in these other species?

R: Thanks for asking this question. We conducted synteny analysis on parrots genomes, and the resulting figure is inserted below. This figure supports the conclusion that chromosome 22 is present in all bird species. In our own analysis, we also double checked that the homologous genes of chicken chromosome 22 are present in all chromosome-level bird genomes that we sampled.

Q: Figure 2a seems to show that the two PARs were assembled on the Z but not on the W. Correct? I think being more explicit about this would help, there is a lot to digest in lines 180-192.

R: Thanks for this comment. Yes, there are two PARs on both the Z and W, one is the original PAR on the ancient sex chromosomes, while the other is a new PAR that was formed on the tips of the added-Z and W chromosomes. However, we only included the PARs on the Z, not on the W, as we present only one haplotype in the genome assembly.

We are sorry about the confusion caused by this previously unclear description. To clarify, we have rewritten lines 180-192: “We found that the added-Z can be divided

into two distinct parts. The sequences proximate to the fusion point, ~3 Mb in size, is hemizygous and exhibits half sequencing depth in both the added-Z and the added-W, resembling the sex determining region (SDR) on ancient sex chromosomes. The remaining sequences (~4Mb), residing at the chromosomal end, are homozygous with autosomal-like sequencing depths (Fig. 1b, Fig. 2a.). Therefore, we hypothesized that this ~4 Mb terminal region is a putative new pseudo autosomal region (new-PAR). Intriguingly, we identified a ~3 Mb inversion between the added-Z and the added-W that coincided with the boundary for the new-PAR (Supplementary Fig. 7). Both breakpoints for the inverted region are located within contigs, suggesting the correct assembly of this inversion (Supplementary Fig. 8). Thus, our analysis supports the role of a structural variation in the formation of a new SDR (Fig. 2d).”

Q: Doesn't PacBio hifi result in haplotype level assemblies? Were the authors unable to assemble the W PAR region?

R: Thanks for this comment. Yes, we were able to assemble two haplotypes of the PARs in the original contigs. But as we addressed above, we included only one haplotype (as we did for autosomes) in the genome assembly and placed the PARs on the Z chromosome. Employing HiFi combined with Hi-C reads allows for generating the local-haplotype assemblies. However, due to a limitation of the methodology, if using the HiFi and Hi-C data to do the whole genomic haplotype level assemblies directly (like an option in the software Hifiasm), the HiFi reads with high sequences similarities, such as long or complicate structured repetitive sequences, from homologous chromosomes may not be correctly phased. Alternatively, for the specially targeted haplotype assembly, an efficient application is the trio-binning, but applying this method requires samples from a two-generation pedigree including parents and offspring to provide a more accurate haplotype genome assembly.

Q: Is the one New-PAR really a PAR as it is typically defined, or is it just the part of chr22 that still recombines like an autosome?

R: Thank you for this important question. In this research, we are unable to look for the synapsis forming by the old-PAR and new-PAR, respectively, during meiosis, as is observed for human PAR1 and PAR2, because we do not have such material from the endangered crested ibis. PARs on sex chromosomes possess a specific higher frequency of recombination than autosomes, in order to ensure the proper progression of meiosis^{5, 6,7,8}. Laurent et al conducted extensive experiments and demonstrated that the unusual recombination in PARs of mice results from the unique dynamic ultrastructure of the PAR, along with cis and trans-regulatory factors⁹. To confirm this mechanism for avian PARs would be challenging and beyond the scope of this study.

In the previous manuscript, we defined the new-PAR only based on bioinformatic observation that the new-PAR region showed autosome-like coverage. To reflect the uncertainty of whether the new-PAR region is a functional PAR, we changed “new-PAR” to putative new-PAR when it is first mentioned in the text.

Q: I imagine the scenario here is that in the past, chr22 fused to the Z and W. Initially,

they would have just recombined like they would have when they weren't fused to the Z and W. Over time, an inversion occurred that limited recombination between parts of chr22, and at that point, those segments of chr22 can start to differentiate from each other. Right? I think this should be made clearer either through a figure or in the text.

R: Thanks. The scenario you described is exactly what we wanted to show and express. Thanks for the helpful suggestion. We have added a figure (Fig. 2d) to illustrate the evolutionary process for the neo-sex chromosomes and also have rewritten this section in the revised manuscript.

Minor.

Q: Lines 14-37. The abstract needs a significant overhaul. There are multiple places where the writing (English) needs to be improved for clarity.

R: Thanks for the suggestion. In the revised version, we have carefully rewritten the abstract.

Q: Lines 40-47. I think it is important that the authors make it very clear that they are just talking about birds and mammals here and that what happens to sex chromosomes can be very different outside of these two groups.

R: Thanks for your valuable suggestion. We have made modifications here to limit our discussion to birds and mammals and avoid misunderstandings. Revised sentence: "In most birds and mammals, the Z and X chromosomes, respectively, are evolutionarily conserved, with relatively stable structures and gene contents^{5, 6}, while the sex-limited W and Y chromosomes exhibit a high degree of heterochromatinization and typically contain only a few genes due to the lack of recombination"

Q: Line 153 and Figure 1e. What is 'rearranged' in this context? Fused?

R: Thanks, we have now changed "rearranged chromosomes" to "chromosomes with inter-chromosomal rearrangements". We have also added a description to the legends of Fig.1: "Rearranged chromosomes in this context refer to the chromosomes with the observed occurrence of fusion events."

Q: Figure 2b. I would consider reworking this figure so that it is clearer how much flanking sequence there is on either side of the putative fusion point and also what is in that picture. Are there any genes? Is this region all repetitive elements?

R: Thank you for the helpful comment. Followed the suggestion, we have redrawn Fig. 2b to display more information about the flanking sequences on both sides of the putative fusion point (100kb on each side) and include the majority of the repetitive elements and the presence of 7 genes. This new figure (Fig. 2b) is provided in the revised manuscript.

Q: Line 182-184 and elsewhere. I would consider changing 'back' and 'front' to something else. This is confusing.

R: Sorry for the unclear writing, we have rewritten this sentence, and changed "front part" into "The sequences proximate to the fusion point", changed "back part"

into “The remaining sequences (~4Mb), residing at the chromosomal end”.

Q: Figure 2c. I would consider adding a chr22 boundary to the figure to help reinforce that the inversion (new-Z/W-SDR) is only a part of the fused chromosome.

R: Thanks for your valuable suggestion, we have added the chromosome 22 boundary to Fig. 2d in the revised manuscript.

Q: The link between sexual selection/sexual antagonism and sex chromosome evolution is not well referenced and also still up for debate (see Ruzicka F, Connallon T. 2020). I think the authors should consider this in the intro and discussion and adjust and cite accordingly.

R: Thank you for the suggestion. We agree that views on sexual selection and sexual antagonism is still an open debate. We have revised our statements and cite Ruzicka F, Connallon T. 2020 in the revised manuscript. We removed sentences that looks like overstatements.

Q: Fig 2b. Do you know where the centromere is located on the Z and W?

R: Thanks for this question. We attempted to annotate the centromere, but unfortunately, without CENP-A ChIP-seq data, we are unsure to identify which satellite DNA are centromeric sequences in crested ibis, in part because the centromeric tandem repeats are not conserved among bird species.

Q: Figure 4a would benefit from having all the birds they are discussing in the text be shown in the synteny comparison.

R: Thanks for this suggestion. We agree with the reviewer's suggestion to expand the display of synteny comparisons for a wider perspective. We have added a synteny comparison between Z and W chromosomes from five different bird species in Supplementary Fig. 15. Only crested ibis shows long and highly conserved synteny blocks (almost the entire length of W). Due to the highly degraded W chromosomes in most bird species (except emus), lineage or species-specific degradations of the W chromosomes was observed in many different species, thus it is hard to have clear and comparable chromosome synteny among the W chromosomes of birds from multiple different lineages.

Q: On a related note, it is confusing how lines between the Neo-z and DNO-Z don't connect segments of the same color. Why are S1 stratum shown with yellow lines connected to green parts of DNO-Z?

R: Thanks for the comment. This is mainly because the paleognathous birds (e.g., emu, DNO) do not share the evolutionary strata with other birds, except for S0. We have added an explanation to the Fig. 4 legend that “The emu strata are demarcated according to Liu *et al.* 2023”. The different colors used in the figure represent the distinct PAR regions (ancient and new) and the evolutionary strata from S0 to S4 of the SDR. As we mentioned above, the extent of W chromosome degeneration in emu is remarkably limited, and the PARs constitute approximately two-thirds of the entire Z

chromosome length¹⁰. In contrast, the crested ibis W chromosome exhibits a more pronounced level of degeneration compared to emu, therefore, the homologous regions of the crested ibis Z-S1 stratum (shown in yellow) remained in an undifferentiated condition on the Z chromosomes of emu (shown in green), which is still a partition of the PAR on the emu sex chromosomes. This explains the linkage of the yellow S1 stratum of the crested ibis Z chromosome with the green of the emu Z chromosome in this figure.

Q: Figure 4b. Add numbers over the bars. You can see broad patterns with the way things are plotted but the numbers would be more informative.

R: Thanks for the suggestion. We have added numbers next to the bars in Fig. 4b.

Reviewer #2:

Recent advances in avian genomics, particularly studies non-model organisms, consistently unveils novel insights into genome organization and evolution. This also applies to the sex chromosomes of birds, which is a subject of manuscript of Xu et al. The evolutionary trajectory of avian sex chromosomes seems to be much more confused and multidirectional than one might think. The present study addresses an important gap in the understanding of sex chromosome evolution in core waterbirds.

R: We greatly appreciate the reviewer's thoughtful feedback and positive remarks.

Using a comprehensive analysis with help of 3rd generation Hi-Fi sequencing methods, the authors revealed two characteristic features of the crested ibis' sex chromosomes: 1) their fusion with the autosome forming neo-sex chromosomes; 2) slow evolutionary rate of W chromosome. The evidences of both features are convincing and the methodology appropriately described. However, the obtained results should be discussed and interpreted more carefully.

R: Thanks for your positive comments. We have revised some detailed explanations and interpretations in the results and discussion sections.

Q: Both features are in line with the theory of sexual selection and support the hypothesis of slower sex chromosome evolution in species with low sexual antagonism. However, this study certainly does not give “a comprehensive understanding of the associations between the sex chromosome evolution and the effects from sex selection or antagonisms” (lines 74-75). Authors highlight this potential connection consistently in the manuscript, exemplified by its placement at the outset of the Abstract. However, their data in no way actually test this association since only one species is analyzed.

R: Thanks for this comment and we agree that the descriptions in lines 74-75 of manuscript were overstated as only one species was analyzed. In the revised manuscript, we changed the statements in this section to “To gain a better understanding of the sex chromosome evolution of birds, we present analyses on the genome of the crested ibis (*Nipponia nippon*), a typical water bird belonging to the Threskiornithidae family of the Pelecaniformes order.” We also rewrote the first sentence of the abstract to “Bird

sex chromosomes play unique roles in sex-determination, and affect the sexual morphology and behavior in bird species.” Additionally, we moderated our statements and cited references of diverse perspectives on the association between sex chromosome evolution and sexual selection.

Q: Moreover, authors do not place these implications to the broader evolutionary context. Indirect data suggest that birds from other taxa may have restricted degradation of W (lines 297-304). Do they possess the same ecological features and patterns of sexual selection?

R: Thanks for this valuable comment. We acknowledge that a broader evolutionary context is needed, in particular, with regards to lines 297-304 in the original manuscript. We have added additional information regarding the ecological characteristics and sexual morphology background of these birds. In the revised manuscript, we rewrote this paragraph as:

“To examine whether the low rate of W degeneration is shared by other core waterbirds, we analyzed the gene content of the W chromosome of the double-crested cormorant (*Phalacrocorax auratus*, PAU), which possesses similar ecological traits with the crested ibis, such as a large body size, monogamous mating system and sexual monomorphism⁶¹. A large number of genes (390 genes) were identified in the double-crested cormorant W chromosome (Fig. 4b).

Moreover, for two other sexually monomorphic and monogamous core waterbird species, the eastern white pelican (*Pelecanus onocrotalus*) and the pygmy cormorant (*Phalacrocorax pygmaeus*), previous cytological experiments showed large-sized W chromosomes. These clues suggest a broader presence of restricted degradation of the W chromosome in large core waterbirds.”

Q: Another feature that is not discussed in this context is the inversion between the neo Z- and W-linked ASCRs, which thus demonstrate pattern typical for W chromosome degradation through recombination suppression. Is it consistent with the slow evolutionary rate of ancient-W?

R: Many thanks for this comment. An inversion is a typical mechanism by which recombination can be suppressed between the Z and W chromosomes, leading to the degradation of the W chromosome. In accordance with the comments from the other reviewer, we redefined the ancient and fused parts of the Z and W chromosomes as “ancient-Z/W” and “added-Z/W” (instead ASCRs), respectively, and the whole sex chromosomes (including the ancient and the added) is still named the neo-sex chromosomes. We discuss the evolutionary rate of the added-W from two aspects: gene loss rate and the gene evolutionary rate.

1. From the perspective of gene content: although the emergence times of S3 and S4 (which were formed by an inversion on the added-W) was speculated to be close based on sequence similarity and dS values, a large number of chromosome 22-homologous genes (80%) were retained on S4 of the added-W, indicating relatively slow degeneration rate of the added-W.

2. From the perspective of gene evolution rate, we conducted a dN/dS analysis to

estimate evolutionary rates of the single-copy orthologous genes on S4 (added-W) and S0-S3 (ancient-W). The results of this test indicate that there is no significant difference in the evolutionary rates of genes between the ancient-W and the added-W, which supports that, similar to the ancient-W genes, genes on the added-W evolve at a low evolutionary rate. The results of both aspects suggest a slow evolutionary rate of added-W similar to that of the ancient-W.

We have supplemented the results as:

“We conducted a *dN/dS* analysis of the orthologous genes, and found that genes from the added-W and ancient-W both have a slow evolutionary rate, and that there was no significant difference in this ratio between the added-W and the ancient-W genes, which supports the conclusion that, similar to the ancient-W genes, genes on the added-W evolved at a slow evolutionary rate (Supplementary Fig.16).

Thus, from the perspective of gene loss rate, and from gene evolutionary rate, these results suggest that, similar to the ancient-W, the added-W genes evolve at a slow rate.”

Q: As for the formation of neo-sex chromosome, the fact that long reads overlap the fusion region is rather reliable proof for the fusion between sex chromosomes and autosomes. However cytological evidence, thus, FISH with chromosome- or region-specific probe, would be still of much value. Perhaps it is worth trying to do this on a related species that does not have endangered status. This would solve two problems – prove the fusion and show that this event is not exclusive for the crested ibis. I would also like to see at least a brief discussion of the mechanisms leading to this unusual, though not unique, fusion.

R: Thanks for your helpful suggestions. During the revision progress, we successfully cultured a cell line from the male crested ibis and designed and synthesized specific probes according to chicken chromosome 22 conserved sequences, which provide the possibility of doing the FISH experiment. As well, to address the second question: “whether other related species also exhibit the same fusion event”, we obtained the sample from a female black-faced spoonbill (*Platalea minor*, PMI, belonging to subfamily Plataeinae of Threskiornithidae) and generated Illumina short read data for contigs assembly. We also conducted a synteny analysis between another ibis, the plumbeous ibis (*Theristicus caerulescens*, TCA) and chicken. Further, we conducted a phylogenetic analysis with the new gene sets, including annotated chr22-orthologous genes from black-faced spoonbill and plumbeous ibis.

The results of these works are as follows:

First, the FISH signals appeared at the crested ibis Z chromosomes, supporting fusions between the sex chromosome and chromosome 22. We added this result as: “Furthermore, FISH experiments using specific probes for chicken chromosome 22 sequences showed that fluorescence signals were present on the pair of Z chromosomes of a crested ibis, which further validates that the neo-sex chromosomes are derived by a fusion event between the ancient sex chromosomes and chromosome 22 (Fig. 2c).”.

Additionally, besides the crested ibis, the result of the synteny analysis supports the same structure for the neo-sex chromosomes (ancient sex chromosomes + chromosome 22) in the plumbeous ibis.

Finally, the phylogenetic analysis results show that the Threskiornithidae gametologs tend to cluster by sex chromosome instead of species.

Given that the crested ibis, plumbeous ibis and black-faced spoonbill, three species, span two different subfamilies within the family Threskiornithidae, this result supports a common origin of the neo-sex chromosome in the earliest ancestor of the Threskiornithidae family.

We have added those sentences in the revised manuscript:

“To investigate whether the neo-sex chromosomes are exclusive for the crested ibis, we carried out a synteny analysis between chicken and another ibis: plumbeous ibis (*Theristicus caerulescens*, TCA). The neo-Z chromosome fused by chr22 and ancient-Z was found in plumbeous ibis, representing a common sex chromosome-autosome fusion event in ibises (Supplementary Fig. 10). Besides, we conducted a phylogenetic analysis of the gametologous genes of the S4 strata, together with their orthologs from various bird species, including the plumbeous ibis and the black-faced spoonbill (*Platalea minor*, PMI), which belongs to the Platalea of Threskiornithidae. The results of this analysis show that Threskiornithidae genes are clustered by chromosomes rather than by species, suggesting that the added-sex chromosomes were likely formed in the common ancestor of the Threskiornithidae (Fig. 2g, Supplementary Fig. 11).

Together, the results of our analyses show that the formation of the neo-sex chromosomes is probably closely associated with the emergence of S3. It is speculated that the neo-sex chromosomes in Threskiornithidae emerged in an early common ancestor, before the first split within this family, but after they diverged from the Ardeidae family.”

Minor comments:

Q: Line 99: Only from the title of this section, one can learn that this assembly of the crested ibis chromosomes is not the first. I think previous attempts should be referred.

R: Thanks for pointing this out. We have referred to the previous draft genome in the revised version, and rewrote the sentence: “The contig N50 reached 16.4 Mb, 630-fold larger than a previous draft genome.”

Line 106: I would classify chromosomes 1-9 as macrochromosomes, and the rest as micro-. The Fig. 1a of the manuscript and the karyotype from Fig. 5 in Wang et al. (2012) clearly show the size gap between chromosomes 9 and 10, which is much larger than between 11 and 12. Moreover, this will be more consistent with the karyotypes of other birds, given the fusions in macrochromosomes in crested ibis.

R: Thanks for this suggestion. In the revised manuscript, we have rewritten line 106 and re-done the comparison of gene density, GC and TE content between the re-defined macro- and micro- chromosomes followed this suggestion. (Revised sentence: “Our assembly anchored more than 95.8% of the contigs onto 29 chromosome models (Fig. 1a), including nine macrochromosomes (chr1-9), 18 microchromosomes (chr10-27) and a pair of ZW sex chromosomes (Supplementary Fig. 1).”)

Q: Line 107-110: First, it is somewhat disappointing that the Hi-Fi reads assembly still does not reach enough resolution to distinguish the smallest microchromosomes. Second, I find this somewhat unfair to claim that "very few non-repetitive genomic sequences are missing from the chromosome assembly" since 6 more microchromosomes did not assemble (which should approximately comprise at least 20-40 Mb based on cytogenetic data). Its importance is probably underestimated given that "micro-chromosomes contained a significantly higher gene density and GC content, but less TEs" especially in light of recent discoveries like Li et al [doi.org/10.1093/molbev/msac066]).

R: Thank for this comment. We share the same disappointment with reviewer about the failure of the HiFi reads in resolving the smallest microchromosomes. According to a recent publication of a complete chicken genome¹¹, perhaps ONT ultra-long will be more useful to assemble the smallest microchromosomes. We have added a clarification in the revised manuscript to highlight this limitation. The added sentence: "The missing chromosomes are probably dot-like microchromosomes that can be better resolved by ONT ultra-long reads."

We are also grateful to the second comment. In the revised manuscript, we have removed this statement.

Q: Line 146-148: I am just wondering in what extent such a comparison of data from comprehensive Hi-Fi assembly with the data obtained by second generation sequencing technology (for which repeats possess a known difficulty) is plausible.

R: Thanks for this insightful comment. We agree that comparison of repeat content between short-read and long-read assemblies could be misleading, therefore, in the revised version of our manuscript, we removed this comparison.

Line 154: Please provide a quantitative characteristic of what you meant by the "chromosomal terminal" (at least in Supplementary).

R: We apologize for the typo and the correct word should be "terminus". We have added this to the legends of Supplementary Fig. 3 that chromosomal terminus means 50 kb regions at the ends of chromosomes.

Line 196: What do you mean claiming that the ancient-PAR "had lengthened by 3.24Mb"? Compared to what?

R: We apologize for the English error. Here, our intention was to state that the total length of the ancient-PAR is 3.24Mb. We have already rewritten this sentence in the revised version as "The ancient-PAR, located at the other tip of the sex chromosomes, (Fig. 2a, d) has a size of 3.24 Mb".

Q: Line 234: I would add an estimated time of neo-sex chromosome formation here.

R: Thanks for this suggestion. Followed this comment, we supplemented the analysis with the two ibises and the black-faced spoonbill (they respectively belong to two different subfamilies within the Threskiornithidae family). The results support the conclusion that the neo-sex chromosomes were caused by the fusion of the ancient-sex

chromosomes and chromosome 22 in the earliest common ancestor of the Threskiornithidae family before the first split within this lineage. The results of gene phylogenetic tree indicated the divergence of S4 stratum in added-Z/W region happened after the lineage divergence of Threskiornithidae and Ardeidae families.

Given the above results, we have added this information in the revised manuscript: “It is speculated that the neo-sex chromosomes in Threskiornithidae emerged in an early common ancestor, before the first split within this family, but after they diverged from the Ardeidae family.”

Line 281-287: I think the phenomenon of higher proportion of multi-copy genes in ancient-W deserves more attention. Are these copies functional or pseudogenes? Do they have active homologues on Z?

R: We thank you for this helpful comment. Multi-copy genes on the W chromosome are not commonly found in birds, and we agree with the reviewer that this phenomenon deserves further attention. We conducted additional analyses on these multi copy genes on these ancient W-SDR and answered the reviewer's questions. These results have been rewritten in the revised version:

“Additionally, all of these genes were observed to have Z-linked gametologs. Fourth, unlike the more than 40% pseudogenized genes on the W linked SDR of ducks, only 10.6% (36/339) of the genes on ancient-W in the crested ibis were pseudogenes, with 21 of these pseudogenes being multi-copy genes (Supplementary Table 4 and 5).”

References cited in the response:

1. Burley JT, Orzechowski SCM, Sin SYW, Edwards SV. Whole-genome phylogeography of the blue-faced honeyeater (*Entomyzon cyanotis*) and discovery and characterization of a neo-Z chromosome. *Mol Ecol* **32**, 1248-1270 (2023).
2. Huang Z, *et al.* Recurrent chromosome reshuffling and the evolution of neo-sex chromosomes in parrots. *Nature Communications* **13**, 944 (2022).
3. Luo H, *et al.* A high-quality genome assembly highlights the evolutionary history of the great bustard (*Otis tarda*, Otidiformes). *Commun Biol* **6**, 746 (2023).
4. Robinson JA, *et al.* Genome-wide diversity in the California condor tracks its prehistoric abundance and decline. *Curr Biol* **31**, 2939-2946.e2935 (2021).
5. Kauppi L, Barchi M, Baudat F, Romanienko PJ, Keeney S, Jasin M. Distinct properties of the XY pseudoautosomal region crucial for male meiosis. *Science* **331**, 916-920 (2011).
6. Soriano P, Keitges EA, Schorderet DF, Harbers K, Gartler SM, Jaenisch R. High rate of recombination and double crossovers in the mouse pseudoautosomal region during male meiosis. *Proc Natl Acad Sci U S A* **84**, 7218-7220 (1987).
7. Lange J, *et al.* The Landscape of Mouse Meiotic Double-Strand Break Formation, Processing, and Repair. *Cell* **167**, 695-708.e616 (2016).
8. Brick K, Smagulova F, Khil P, Camerini-Otero RD, Petukhova GV. Genetic recombination is directed away from functional genomic elements in mice. *Nature* **485**, 642-645 (2012).
9. Acquaviva L, *et al.* Ensuring meiotic DNA break formation in the mouse pseudoautosomal region. *Nature* **582**, 426-431 (2020).
10. Liu J, *et al.* A new emu genome illuminates the evolution of genome configuration and

- nuclear architecture of avian chromosomes. *Genome Research* **31**, 497-511 (2021).
11. Huang Z, *et al.* Evolutionary analysis of a complete chicken genome. *Proc Natl Acad Sci U S A* **120**, e2216641120 (2023).

REVIEWERS' COMMENTS

Reviewer #1 (Remarks to the Author):

Xu et al have resubmitted a manuscript on some comparative genomics work they have done using a new crested ibis genome assembly that includes an interesting neo-sex chromosome. I reviewed the first draft of this paper and find this version to be much improved. The authors have attempted to answer all the comments/critiques from the first review and they have even added additional data in the form of Illumina reads from a different species, and some FISH experiments to try to confirm the sex chromosome fusion. I have no remaining concerns with the manuscript although I do have two very minor things that I want to mention.

1) I would consider changing the color or adding some additional text to Fig2d. Having chr22 be a light green doesn't differentiate it enough from ancient par and I felt like I still needed to stare at this figure for too long to understand the evolutionary model the authors are proposing.

2) I'm still a little confused about why the authors are unable to guess as to where the centromeres are in this assembly. It is not super important for their paper so it can be left for another paper/analysis, but do we know these are metacentric chromosomes (like what is shown in 2d)? Do chromosome squashes suggest this? Just a bit surprised because the Hi-C plot (Fig 2a) doesn't seem to show the typical signatures of a metacentric chromosomes (the Z or the W).

Reviewer #2 (Remarks to the Author):

The authors has addressed issues raised in the reviews and thoroughly revised the paper by conducting new experiments (confirming of chromosomal fusion by FISH, sequencing of the related species), adding more data (sequencing coverage, synteny analysis etc), and discussing the results more carefully (avoiding overstatements and using terminology consistently).

However, the text still requires careful stylistic and grammatical revision.

Here are just a few typos, which are clearly noticeable:

- Line 29 in the Abstract: authors apparently have meant "Neo-W chromosome geneS"
- Line 69: challanged  challenged
- Lines 109-111: the sentence needs to be revised
- Lines 188-189: check the grammar
- Y-axis of Fig.1b: Maltes  of Males
- Line 961 - boxex  of boxes
- Lime 961 - do you mean hybridization signals?

RESPONSE TO REVIEWERS' COMMENTS

Reviewer #1 (Remarks to the Author):

Xu et al have resubmitted a manuscript on some comparative genomics work they have done using a new crested ibis genome assembly that includes an interesting neo-sex chromosome. I reviewed the first draft of this paper and find this version to be much improved. The authors have attempted to answer all the comments/critiques from the first review and they have even added additional data in the form of Illumina reads from a different species, and some FISH experiments to try to confirm the sex chromosome fusion. I have no remaining concerns with the manuscript although I do have two very minor things that I want to mention.

1) I would consider changing the color or adding some additional text to Fig2d. Having chr22 be a light green doesn't differentiate it enough from ancient par and I felt like I still needed to stare at this figure for too long to understand the evolutionary model the authors are proposing.

2) I'm still a little confused about why the authors are unable to guess as to where the centromeres are in this assembly. It is not super important for their paper so it can be left for another paper/analysis, but do we know these are metacentric chromosomes (like what is shown in 2d)? Do chromosome squashes suggest this? Just a bit surprised because the Hi-C plot (Fig 2a) doesn't seem to show the typical signatures of a metacentric chromosomes (the Z or the W).

Reviewer #2 (Remarks to the Author):

The authors has addressed issues raised in the reviews and thoroughly revised the paper by conducting new experiments (confirming of chromosomal fusion by FISH, sequencing of the related species), adding more data (sequencing coverage, synteny analysis etc), and discussing the results more carefully (avoiding overstatements and using terminology consistently).

However, the text still requires careful stylistic and grammatical revision.

Here are just a few typos, which are clearly noticeable:

- Line 29 in the Abstract: authors apparently have meant "Neo-W chromosome geneS"
- Line 69: challanged  challenged
- Lines 109-111: the sentence needs to be revised
- Lines 188-189: check the grammar
- Y-axis of Fig.1b: Maltes  of Males
- Line 961 - boxex  of boxes
- Lime 961 - do you mean hybridization signals?

REVIEWERS' COMMENTS

Reviewer #1 (Remarks to the Author):

Xu et al have resubmitted a manuscript on some comparative genomics work they have done using a new crested ibis genome assembly that includes an interesting neo-sex chromosome. I reviewed the first draft of this paper and find this version to be much improved. The authors have attempted to answer all the comments/critiques from the first review and they have even added additional data in the form of Illumina reads from a different species, and some FISH experiments to try to confirm the sex chromosome fusion. I have no remaining concerns with the manuscript although I do have two very minor things that I want to mention.

R: Thanks for your kind comments and we appreciate your review work.

1) I would consider changing the color or adding some additional text to Fig2d. Having chr22 be a light green doesn't differentiate it enough from ancient par and I felt like I still needed to stare at this figure for too long to understand the evolutionary model the authors are proposing.

R: Thanks for your suggestion. We have changed the color and added additional legends to Fig2d.

2) I'm still a little confused about why the authors are unable to guess as to where the centromeres are in this assembly. It is not super important for their paper so it can be left for another paper/analysis, but do we know these are metacentric chromosomes (like what is shown in 2d)? Do chromosome squashes suggest this? Just a bit surprised because the Hi-C plot (Fig 2a) doesn't seem to show the typical signatures of a metacentric chromosomes (the Z or the W).

R: Thanks. The DNA sequences of centromeres in avian species are not conserved. Without CENP-A ChIP-seq data, it is challenging to definitively identify which tandem repeats in the genome correspond to centromeric sequences. Hi-C maps are also inadequate for determining centromeric positions. However, even though centromeric sequences cannot be directly annotated from the genome in the absence of CENP-A ChIP-seq data, G-banding analysis of cytological experiments on the crested ibis indicates that the Z chromosome is a median centromere chromosome and the W chromosome features a sub-terminal centromere¹. Therefore, the models of Z and W chromosomes in Figure 2d were drawn based on the results of G-banding analysis.

Reviewer #2 (Remarks to the Author):

The authors has addressed issues raised in the reviews and thoroughly revised the paper by conducting new experiments (confirming of chromosomal fusion by FISH, sequencing of the related species), adding more data (sequencing coverage, synteny analysis etc), and discussing the results more carefully (avoiding overstatements and using terminology consistently).

R: Thanks for your kind comments.

However, the text still requires careful stylistic and grammatical revision.

R: Thanks for your advice. We have carefully reviewed and revised the manuscript to address the stylistic and grammatical concerns.

Here are just a few typos, which are clearly noticeable:

- Line 29 in the Abstract: authors apparently have meant "Neo-W chromosome genes"

R: Thanks for pointing it out. We have changed "gene" to "genes".

- Line 69: challenged  challenged

R: Thank you very much, we have corrected this error in writing.

- Lines 109-111: the sentence needs to be revised.

R: Thanks, we have revised this sentence as: "We identified the Z and W chromosomes by comparing the sequencing coverage between males and females. In females, both the Z and W chromosomes exhibit half the sequencing coverage of autosomes, while in males, the W chromosome displays sparse coverage".

- Lines 188-189: check the grammar.

R: Thanks, we have checked the grammar and revised.

- Y-axis of Fig.1b: Maltes  of Males

R: Thanks for pointing it out. Revised.

- Line 961 - boxex  of boxes

R: Thanks for pointing it out. Revised.

- Line 961 - do you mean hybridization signals?

R: Yes. We have revised this sentence as: "FISH images for the probes of chicken chr22 hybridized in crested ibis chrZ."

Reference cited in this response

- 1 Wang J, et al. Establishment of crested ibis cell line and observation of its biological characteristics. *Zoological Research* 33, 591-596 (2012).